# Does Knowledge Distillation Really Work?

**Samuel Stanton**
NYU

**Pavel Izmailov**
NYU

**Polina Kirichenko**
NYU

**Alexander A. Alemi**
Google Research

**Andrew Gordon Wilson**
NYU

## Abstract

Knowledge distillation is a popular technique for training a small student network to emulate a larger teacher model, such as an ensemble of networks. We show that while knowledge distillation can improve student generalization, it does not typically work as it is commonly understood: there often remains a surprisingly large discrepancy between the predictive distributions of the teacher and the student, even in cases when the student has the capacity to perfectly match the teacher. We identify difficulties in optimization as a key reason for why the student is unable to match the teacher. We also show how the details of the dataset used for distillation play a role in how closely the student matches the teacher — and that more closely matching the teacher paradoxically does not always lead to better student generalization.

## 1 Introduction

Large, deep networks can learn representations that generalize well. While smaller, more efficient networks lack the *inductive biases* to find these representations from training data alone, they may have the *capacity* to represent these solutions [e.g., 2, 18, 32, 45]. Influential work on *knowledge distillation* [22] argues that Bucilǎ et al. [5] "demonstrate convincingly that the knowledge acquired by a large ensemble of models [the teacher] can be transferred to a single small model [the student]". Indeed this quote encapsulates the conventional narrative of knowledge distillation: a student model learns a high-fidelity representation of a larger teacher, enabled by the teacher's soft labels.

Conversely, in Figure 1 we show that with modern architectures knowledge distillation can lead to students with very different predictions from their teachers, even when the student has the capacity to perfectly match the teacher. Indeed, it is becoming well-known that in self-distillation the student fails to match the teacher and, paradoxically, student generalization improves as a result [14, 40]. However, when the teacher is a large model (e.g. a deep ensemble) improvements in fidelity translate into improvements in generalization, as we show in Figure 1(b). For these large models there is still a significant accuracy gap between student and teacher, so fidelity is aligned with generalization.

We will distinguish between *fidelity*, the ability of a student to match a teacher's predictions, and *generalization*, the performance of a student in predicting unseen, in-distribution data. We show that in many cases it is surprisingly difficult to obtain good student fidelity. In Section 5 we investigate the hypothesis that low fidelity is an *identifiability* problem that can be solved by augmenting the distillation dataset. In Section 6 we investigate the hypothesis that low fidelity is an *optimization* problem resulting in a failure of the student to match the teacher even on the original training dataset. We present a summary of our conclusions in Section 7.

*Does knowledge distillation really work?* In short: *Yes*, in the sense that it often improves student generalization. *No*, in that knowledge distillation often fails to live up to its name, transferring very limited knowledge from teacher to student.

35th Conference on Neural Information Processing Systems (NeurIPS 2021).

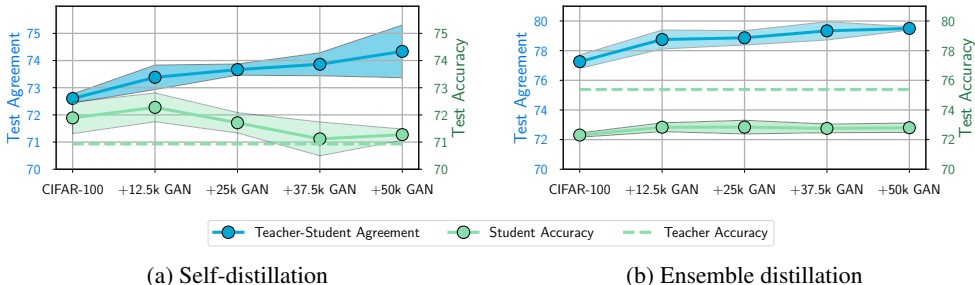

(a) Self-distillation        (b) Ensemble distillation

Figure 1: **Evaluating the fidelity of knowledge distillation.** The effect of enlarging the CIFAR-100 distillation dataset with GAN-generated samples. **(a)**: The student and teacher are both single ResNet-56 networks. Student fidelity increases as the dataset grows, but test accuracy decreases. **(b)**: The student is a single ResNet-56 network and the teacher is a 3-component ensemble. Student fidelity again increases as the dataset grows, but test accuracy now slightly increases. The shaded region corresponds to $\mu \pm \sigma$, estimated over 3 trials.

## 2 Related Work

Knowledge distillation can improve model efficiency [38, 45], unsupervised domain adaptation [37], improved object detection [9], model transparency [48], and adversarial robustness [15, 42].

Seminal work by Bucilă et al. [5] showed that teacher-ensembles with thousands of simple components could be compressed into a single shallow network that matched or outperformed its teacher. Other early work proposed distilling ensembles of shallow networks into a single network [55], an idea which resonates with more recent work on the distillation of deep ensembles [2, 7, 46, 50, 53]. Recently Fakoor et al. [13] developed a data-augmentation scheme for the distillation of large ensembles of simple models for tabular data, achieving impressive results on a wide range of tabular benchmarks. Malinin et al. [35] proposed a method to model the implicit distribution over predictive distributions from which the ensemble component predictive distributions are drawn, rather than just the ensemble model average.

Our work focuses explicitly on student fidelity, decoupling our understanding of good fidelity from good generalization. We show that achieving good fidelity is extremely difficult, even with a variety of interventions, and seek to *understand*, by systematically considering several hypotheses, why knowledge distillation does not produce high fidelity students for modern architectures and datasets. In contrast, the distillation literature focuses largely on improving student generalization, without particularly distinguishing between fidelity and generalization.

For example, concurrent work by Beyer et al. [4] does not carefully distinguish generalization and fidelity metrics, but they assert that high student fidelity is *conceptually* desirable and apparently difficult to achieve when measured as the gap between teacher and student accuracy. As a result their work focuses most heavily on practical modifications to the distillation procedure for the best student top-1 accuracy. In this paper we investigate many of the same prescriptions, including careful treatment of data augmentation (such as showing the teacher and student the exact same input images), the addition of MixUp, and extended training duration. We also find that such interventions do improve student accuracy, but there still remains a large discrepancy between the predictive distributions of the teacher and the student. We also investigate multiple optimizers. While we do not pursue Shampoo [17, 1] specifically, Beyer et al. [4] find similar qualitative results for Shampoo and Adam, besides faster convergence for Shampoo.

## 3 Preliminaries

We will focus on the supervised classification setting, with input space $\mathcal{X}$ and label space $\mathcal{Y}$, where $|\mathcal{Y}| = c$. Let $f : \mathcal{X} \times \Theta \to \mathbb{R}^c$ be a classifier parameterized by $\theta \in \Theta$ whose outputs define a categorical predictive distribution over $\mathcal{Y}$, $\hat{p}(y = i|\mathbf{x}) = \sigma_i(f(\mathbf{x}, \theta))$, where $\sigma_i(\mathbf{z}) := \exp(z_i)/\sum_j \exp(z_j)$ is the softmax link function. We will often refer to the outputs of a classifier $\mathbf{z} := f(\mathbf{x}, \theta)$ as *logits*. For convenience, we will use $t$ and $s$ as shorthand for $f_{\text{teacher}}$ and $f_{\text{student}}$, respectively. When the

teacher is an $m$-component ensemble, the component logits $(\mathbf{z}_1, \ldots, \mathbf{z}_m)$, where $\mathbf{z}_i = f_i(\mathbf{x}, \theta_i)$, are combined to form the teacher logits: $\mathbf{z}_t = \log\left(\sum_{i=1}^m \sigma(\mathbf{z}_i)/m\right)$. These combined logits correspond to the predictive distribution of the ensemble model average. The experiments in the main text consider $m \in \{1, 3, 5\}$, and we include results up to $m = 12$ in Appendix B.2.[1]

### 3.1 Knowledge Distillation

Hinton et al. [22] proposed a simple approach to knowledge distillation. The student minimizes a weighted combination of two objectives, $\mathcal{L}_s := \alpha \mathcal{L}_{\text{NLL}} + (1-\alpha)\mathcal{L}_{\text{KD}}$, where $\alpha \in [0, 1)$. Specifically,

$$\mathcal{L}_{\text{NLL}}(\mathbf{z}_s, \mathbf{y}) := -\sum_{j=1}^c y_j \log \sigma_j(\mathbf{z}_s), \quad \mathcal{L}_{\text{KD}}(\mathbf{z}_s, \mathbf{z}_t) := -\tau^2 \sum_{j=1}^c \sigma_j\left(\frac{\mathbf{z}_t}{\tau}\right) \log \sigma_j\left(\frac{\mathbf{z}_s}{\tau}\right). \quad (1)$$

$\mathcal{L}_{\text{NLL}}$ is the usual supervised cross-entropy between the student logits $\mathbf{z}_s$ and the one-hot labels $\mathbf{y}$. Recalling that $\text{KL}(p\|q) = \sum_j p_j(\log q_j - \log p_j)$, we see that $\mathcal{L}_{\text{NLL}}$ is equivalent (up to a constant) to the KL from the empirical data distribution to the student predictive distribution ($\hat{p}_s$). $\mathcal{L}_{\text{KD}}$ is the added knowledge distillation term that encourages the student to match the teacher. It is the cross-entropy between the teacher and student predictive distributions $\hat{p}_t = \sigma(\mathbf{z}_t)$ and $\hat{p}_s = \sigma(\mathbf{z}_s)$, *both* scaled by a temperature hyperparameter $\tau > 0$. If $\tau = 1$ then $\mathcal{L}_{\text{KD}}$ is similarly equivalent to the KL from the teacher to the student, $\text{KL}(\hat{p}_t\|\hat{p}_s)$. Since we focus on distillation fidelity, we choose $\alpha = 0$ for all experiments in the main text to avoid any confounding from true labels, but we also include a limited ablation of $\alpha$ in Figure 14 in Appendix C.5 for the curious reader.

As $\tau \to +\infty$, $\nabla_{\mathbf{z}_s} \mathcal{L}_{\text{KD}}(\mathbf{z}_s, \mathbf{z}_t) \approx \mathbf{z}_t - \mathbf{z}_s$, and thus in the limit $\nabla_{\mathbf{z}_s} \mathcal{L}_{\text{KD}}$ is approximately equivalent to $\nabla_{\mathbf{z}_s} \|\mathbf{z}_t - \mathbf{z}_s\|_2^2 / 2$, assigning equal significance to every class logit, regardless of its contribution to the predictive distribution. In other words $\tau$ determines the "softness" of the teacher labels, which in turn determines the allocation of student capacity. If the student is much smaller than the teacher, the student capacity can be focused on matching the teacher's top-$k$ predictions, rather than matching the full teacher distribution by choosing a moderate value (e.g. $\tau = 4$). In Appendix B.1 we include further discussion on the interplay of teacher ensemble size, teacher network capacity, and distillation temperature on the student labels.

The teacher and student often share at least some training data. It is also common to enlarge the student training data in some way (e.g. incorporating unlabeled examples as in Ba and Caruana [2]). When there is a possibility of confusion, we will refer to the student's training data as the *distillation data* to distinguish it from the teacher's training data.

### 3.2 Metrics and Evaluation

To measure generalization, we report top-1 accuracy, negative log-likelihood (NLL) and expected calibration error (ECE) [16]. To measure fidelity, we report the following:

$$\text{Average Top-1 Agreement} := \frac{1}{n}\sum_{i=1}^n \mathbb{1}\{\arg\max_j \sigma_j(\mathbf{z}_{t,i}) = \arg\max_j \sigma_j(\mathbf{z}_{s,i})\}, \quad (2)$$

$$\text{Average Predictive KL} := \frac{1}{n}\sum_{i=1}^n \text{KL}\left(\hat{p}_t(\mathbf{y}|\mathbf{x}_i) \,\|\, \hat{p}_s(\mathbf{y}|\mathbf{x}_i)\right), \quad (3)$$

Eqn. (2) is the average *agreement* between the student and teacher's top-1 label. Eqn. (3) is the average KL divergence from the predictive distribution of the teacher to that of the student, a measure of fidelity sensitive to all of the labels.

While improvements in generalization metrics are relatively easy to understand, interpreting fidelity metrics requires some care. For example, suppose we have three independent models: $f_1$, $f_2$, and $f_3$ that respectively achieve 55%, 75%, and 95% test accuracy. $f_1$ and $f_3$ can agree on at most 60% of points, whereas $f_2$ and $f_3$ agree on at least 70%, but it would obviously be incorrect to make any claim about $f_2$ being a better distillation of $f_3$ since each model was trained completely independently. To account for such confounding when evaluating the distillation of a student $s$ from a teacher $t$, we also evaluate another student $s'$ distilled through an identical procedure from an independent teacher.

---

[1]Code for all experiments can be found here: https://github.com/samuelstanton/gnosis.

By comparing the fidelity of $(t, s)$ and $(t, s')$ we can distinguish between a generic improvement in generalization and an improvement specifically to fidelity. If $s$ and $s'$ have comparable fidelity, then the students agree with the teacher at many points because they generalize well, and not the reverse.

## 4 Knowledge Distillation Transfers Knowledge Poorly

In this section, we present evidence that we are not able to distill large networks such as a ResNet-56 with high fidelity, and discuss why high fidelity is an important objective.

### 4.1 When is knowledge transfer successful?

We first consider the easy task of distilling a LeNet-5 teacher into an identical student network as a motivating example. We train the teacher on a random subset of 200 examples from the MNIST training set for 100 epochs, resulting in a $84\%$ to $86\%$ teacher test accuracy across different subsets.[2] We then distill the teacher using the full MNIST train dataset with 60,000 examples, as well as 25%, 50%, and 100% of the EMNIST train dataset [11]. The EMNIST train set contains 697,932 images.

In Figure 2 we see that knowledge distillation works as expected. With enough examples the student learns to make the same predictions as the teacher (over 99% top-1 test agreement). Notably, in this case, self-distillation does not *improve* generalization, since the slight difference between the teacher and student accuracy is explained by variance between trials.

Now we consider a more challenging task: distilling a ResNet-56 teacher trained on CIFAR-100 into an identical student network (Figure 1, left). Since no dataset drawn from the same distribution as CIFAR-100 is publicly available, to augment the distillation data, we instead combined samples from an SN-GAN [39] pre-trained on CIFAR-100 with the original CIFAR-100 train dataset. Appendix A.3 details the hyperparameters and training procedure for the GAN, teacher, and student.

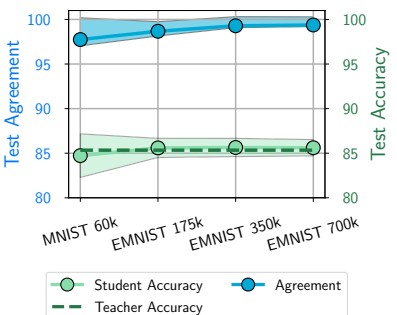

Figure 2: LeNet-5 self-distillation on MNIST with additional distillation data. The shaded region corresponds to $\mu \pm \sigma$, estimated over 3 trials.

Like the MNIST experiment, as we enlarge the distillation dataset the student fidelity improves. However, in this case the improvement is modest, with the fidelity reaching nowhere near 99% test agreement. Since a ResNet-56 has many more parameters than a LeNet-5, it is possible that the student simply has not seen enough examples to perfectly emulate the teacher, a hypothesis we discuss in more detail in Section 5.1. Also, like the MNIST experiment, as the distillation dataset grows the student accuracy approaches the teacher's. *Unlike* the MNIST experiment, the student test accuracy is higher than the teacher's when the distillation dataset is small, so increasing fidelity *decreases* student generalization.

### 4.2 What can self-distillation tell us about knowledge distillation in general?

We have seen in Figure 1(a) that with self-distillation the student can exceed the teacher performance, in accordance with Furlanello et al. [14]. This result is only possible by virtue of failing at the distillation procedure: if the student matched the teacher perfectly then the student could not outperform the teacher. On the other hand, if the teacher generalizes significantly better than an independently trained student, we would expect the benefits of fidelity to dominate other regularization effects associated with not matching the teacher. This setting reflects the original motivation for knowledge distillation, where we wish to faithfully transfer the representation discovered by a large model or ensemble of models into a more efficient student.

In Figure 1(b) we see that if we move from self-distillation to the distillation of a 3 ResNet-56 teacher ensemble, fidelity becomes positively correlated with generalization. But there is still a significant

---

[2]We took only a subset of the MNIST train set since otherwise every teacher network as well as the ensemble would achieve over 99% test accuracy.

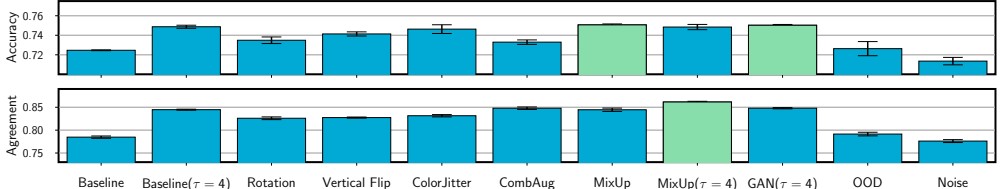

Figure 3: **Data augmentation and distillation**: Test accuracy and teacher-student agreement when distilling a 5-component ResNet-56 teacher ensemble into a ResNet-56 student on CIFAR-100 with varying augmentation policies. The best performing policy is shown in green, results averaged over 3 runs. Additional metrics are reported in Figure 11 in Appendix C. Mixup and GAN augmentation provide the best generalization, and Mixup($\tau = 4$) provides the best fidelity. The baseline policy (crops and flips) with $\tau = 4$ is a surprisingly strong baseline. The error bars indicate $\pm\sigma$.

gap in fidelity, even after the distillation set is enlarged with $50k$ GAN samples. In practice, the gap remains large enough that higher fidelity students do not always have better generalization, and the regularization effects we see in self-distillation do play a role for more broadly understanding student generalization. We will indeed show in Section 5 that higher fidelity students do not always generalize better, even if the teacher generalizes much better than the student.

### 4.3 If distillation already improves generalization, why care about fidelity?

While knowledge distillation does often improve generalization, understanding the relationship between fidelity and generalization, and how to maximize fidelity, is important for several reasons — including better generalization!

**Better generalization in distilling large teacher models and ensembles.** Knowledge distillation was initially motivated as a means to deploy powerful models to small devices or low-latency controllers [e.g., 10, 21, 26, 52, 54]. While in self-distillation generalization and fidelity are in tension, there is often a significant disparity in generalization between large teacher models, including ensembles, and smaller students. We have seen this disparity in Figure 1(b). We additionally show in Figure 10 in Appendix B.1 that as we increase the number of ensemble components, the generalization disparity between teacher and distilled student increases. Improving student fidelity is the most obvious way to close the generalization disparity between student and teacher in these settings. Even if one exclusively cares about student accuracy, fidelity is a key consideration outside self-distillation.

**Interpretability and reliability.** Knowledge distillation has been identified as a means to *transfer representations* discovered by large black-box models into simpler more interpretable models, for example to provide insights into medical diagnostics, or discovering rules for understanding sentiment in text [e.g., 23, 24, 6, 33, 8]. The ability to perform this transfer could have extraordinary scientific consequences: large models can often discover structure in data that we would not have anticipated a priori. Moreover, we often want to transfer properties such as well-calibrated uncertainties or robustness, which have been well-established for larger models, so that we can safely deploy more efficient models in their place. In both cases, achieving good distillation fidelity is crucial.

**Understanding.** The name *knowledge distillation* implies we are transferring knowledge from the teacher to the student. For this reason, improved student generalization as a consequence of a distillation procedure is sometimes conflated with fidelity. Decoupling fidelity and generalization, and explicitly studying fidelity, is foundational to understanding how knowledge distillation works and how we can make it more useful across a variety of applications.

### 4.4 Possible causes of low distillation fidelity

If we are able to match the student model to the teacher on a comprehensive distillation dataset, we expect it to match on the test data as well, achieving high distillation fidelity[3]. Possible causes of the poor distillation fidelity in our CIFAR-100 experiments include:

---

[3]See, for example, Lemma 1 in Fakoor et al. [13].

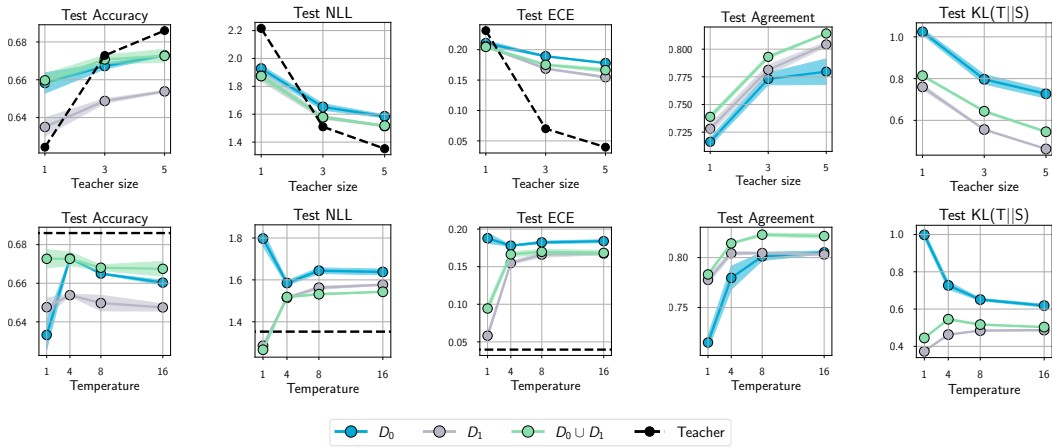

Figure 4: **Data recycling and distillation**: results on subsampled CIFAR-100. **Top:** We fix the temperature ($\tau = 4$) and vary the number of ensemble components ($m$), comparing students distilled on the same dataset as the teacher ($\mathcal{D}_0/\mathcal{D}_0$), a reserved dataset ($\mathcal{D}_0/\mathcal{D}_1$), or both ($\mathcal{D}_0/\mathcal{D}_0 \cup \mathcal{D}_1$). Distilling on both produces the best result, while distilling on $\mathcal{D}_0$ increases accuracy and decreases fidelity, relative to $\mathcal{D}_1$. **Bottom:** We repeat the experiment, but fix $m = 3$ and vary $\tau$. The shaded region corresponds to $\mu \pm \sigma$, estimated over 3 trials.

**Student capacity** – We observe low fidelity even in the self-distillation setting, so we can rule out student capacity as a primary cause, but we also confirm in Figure 12 in Appendix C.1 that increasing the student capacity has very little effect on fidelity in the ensemble-distillation setting.

**Network architecture** – Low fidelity could be specific to ResNet-like architectures, an explanation we rule out by showing similar results with VGG networks [47] in Figure 13 in Appendix C.2.

**Dataset scale and complexity** – we provide similar results in Section C.3 for ImageNet, showing that our findings apply to datasets of larger scale and complexity.

**Data domain** – Similarly in Section C.4 we observe low distillation fidelity in the context of text classification (sentiment analysis on the IMDB dataset), showing our results are relevant beyond image classification.

**Identifiability** (Section 5) – the distillation data is insufficient to distinguish high-fidelity and low-fidelity students. In other words, matching the teacher predictions on the distillation dataset does not lead to matching predictions on the test data.

**Optimization** (Section 6) – we are unable to solve the distillation optimization problem sufficiently well. The student does not agree with the teacher on test because it does not even agree on train.

## 5 Identifiability: Are We Using the Right Distillation Dataset?

We investigate whether it is possible to attain the level of fidelity observed with LeNet-5s on MNIST with ResNets on CIFAR-100 by addressing the *identifiability* problem — have we shown the student enough of the right input-teacher label pairs to define the solution we want?

### 5.1 Should we do more data augmentation?

Data augmentation is a simple and practical method to increase the support of the distillation data distribution. If identifiability is a primary cause of poor distillation fidelity, using a more extensive data augmentation strategy during distillation should improve fidelity.

To test this hypothesis, we evaluated the effect of several augmentation strategies on student fidelity and generalization. In Figure 3, the teacher is a 5-component ensemble of ResNet-56 networks trained on CIFAR-100 with the *Baseline* augmentation strategy: horizontal flips and random crops.

We report the student accuracy and teacher-student agreement for each augmentation strategy, and also include results for *Baseline* with $\tau = 1$ and $\tau = 4$ to demonstrate the effect of logit tempering.

We first observe that the best augmentation policies for generalization, *MixUp*, and *GAN*[4], are not the best policies for fidelity. Furthermore, although many augmentation strategies enable slightly higher distillation fidelity compared to *Baseline ($\tau = 1$)*, even the best augmentation policy, *Mixup ($\tau = 4$)*, only achieves a modest 86% test agreement. In fact the *Baseline ($\tau = 4$)* policy is quite competitive, achieving $84.5\%$ test agreement. Many of the augmentation strategies also slightly improve teacher-student KL relative to *Baseline ($\tau = 4$)* (see Figure 11).

In Figure 11 in Appendix B.3 we report all generalization and fidelity metrics for a range of ensemble sizes, as well as the results for the independent student baseline discussed in Section 3.2. Often these independent students, taught how to mimic a completely different model, have nearly as good test agreement with the teacher as the student explicitly trained to emulate it. See Appendix A.1 for a detailed description of the augmentation procedures.

**Should data augmentation be close to the data distribution?**    In theory, *any* data augmentation should help with identifiability: if a student matches a teacher on more data, it is more likely to match the teacher elsewhere. However, the *Noise* and *OOD* augmentation strategies based on noise and out-of-distribution data fail on all metrics, decreasing performance compared to the baseline. In practice, data augmentation has an effect beyond improving identifiability — it has a regularizing effect, making optimization more challenging. We explore this facet of data augmentation in Section 6.

The slight improvements to fidelity with extensive augmentations suggest that increasing the support of the distillation dataset can indeed improve distillation fidelity. However, since the benefit is so small compared to heuristics like logit tempering (which does not modify the support at all), it is very unlikely that an insufficient quantity of teacher labels is the primary obstacle to high fidelity.

## 5.2    The data recycling hypothesis

If simply showing the student *more* labels does not always significantly improve fidelity, perhaps we are not showing the student the *right* labels. Additional data augmentation during distillation does give the student more teacher labels to match, but also introduces a distribution shift between the images the teacher was trained on and the images the student is distilling on. Even when the teacher and student have the same augmentation policy, reusing the teacher's training data for distillation violates the assumptions of empirical risk minimization (ERM) because the distillation data is *not* an independent draw from the true joint distribution over images and teacher labels. What if there was no augmentation distribution shift, and the student was distilled on a fresh draw from the joint test distribution over images and teacher labels?

To investigate the effect of recycling teacher data during distillation we randomly split the CIFAR-100 training dataset $\mathcal{D}$ into two equal parts, $\mathcal{D}_0$ and $\mathcal{D}_1$. We train teacher ResNet-56 ensembles on $\mathcal{D}_0$, and then compare $s_0$, a student distilled on the original $\mathcal{D}_0$, $s_1$, a student distilled on the unseen $\mathcal{D}_1$, and $s_{0 \cup 1}$, a student distilled on both: $\mathcal{D}_0 \cup \mathcal{D}_1$. Note that the students cannot access the true labels, only those provided by the teacher. We present the results in Figure 4, varying the ensemble size in the top row and the logit temperature in the bottom row.

Surprisingly, $s_0$ attains higher test accuracy than $s_1$, while showing worse ECE and lower fidelity (measured by test teacher-student agreement and test teacher-student KL). Therefore, the hypothesis that $s_1$ should be a higher fidelity distillation of the teacher than $s_0$ does hold, but the gain in fidelity *does not* result in $s_1$ best replicating the teacher's accuracy. The best attributes of $s_0$ and $s_1$ are combined by $s_{0 \cup 1}$, which coincides with how unlabeled data is typically used in practice [2]. The reason for this puzzling observation is simply that for the larger teachers fidelity has not improved *enough* to also improve generalization. In fact, the best teacher-student agreement is only around $85\%$, no improvement when compared to the results from extensive data augmentation in the last section. We again find that modifying the distillation data can slightly improve fidelity, but the evidence does not support blaming poor distillation fidelity on the wrong choice of distillation data.

---

[4]Unlike Figure 1, for Figure 3 we generated new GAN samples every epoch, to mimic data augmentation.

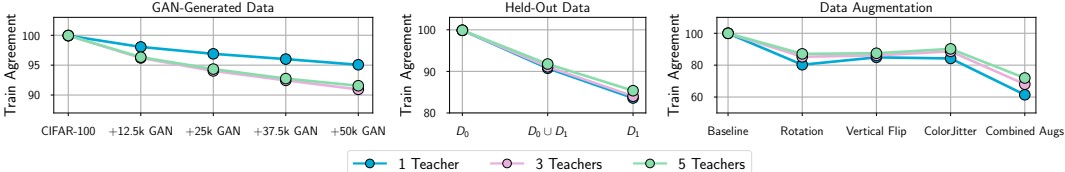

Figure 5: The train agreement for teacher ensembles ($m \in \{1, 3, 5\}$) and student on the distillation data for a ResNet-56 on CIFAR-100 under different augmentation policies. In all panels, increasing the softness of the teacher labels by adding examples not in the teacher train data makes distillation more difficult. **Left**: agreement for the synthetic GAN-augmentation policy from Figure 1. **Middle**: agreement from subsampled CIFAR-100 experiment in Figure 4. **Right**: agreement for some of the augmentation policies in Figure 3. The shaded region is not visible because the variance is very low.

# 6 Optimization: Does the Student Match the Teacher on Distillation Data?

If poor fidelity is not primarily an identifiability problem from the wrong choice of distillation data, perhaps there is a simpler explanation. Up to this point, we have focused on student fidelity on a held-out test set. Now we turn our attention to student behavior on the distillation data itself. Does the student match the teacher on the data it is trained to match it on?

## 6.1 More distillation data lowers train agreement

In Figure 1 we presented an experiment distilling ResNet-56 networks on CIFAR-100 augmented with synthetic GAN-generated images. We saw that enlarging the distillation dataset leads to improved teacher-student agreement on test, but the agreement remains relatively low (below $80\%$) even for the largest distillation dataset that we considered. In Figure 5 (left panel), we report the teacher-student agreement for the same experiment, but now on the distillation dataset. We now observe the opposite trend: as the distillation dataset becomes larger, it becomes more challenging for the student to match the teacher. Even when the student has identical capacity to the teacher, the student only achieves $95\%$ agreement with the teacher when we use $50k$ synthetic images for distillation.

The drop in train agreement is even more pronounced when we use extensive data augmentation. In Figure 5, right panel, we report the teacher-student agreement on the train set with data augmentation for a subset of augmentation strategies presented in Section 5.1. We use the CIFAR-100 dataset and the ResNet-56 model for the teachers and the students (for details, see Section 5.1). In each case, we measure agreement on the augmented training set that was used during distillation. While for the baseline augmentation strategy, we can achieve almost perfect teacher-student agreement, for heavier augmentations the agreement drops dramatically. For the *Rotation*, *Vertical Flip* and *Color Jitter* augmentations, the agreement is between $80\%$ and $90\%$ for all the considered teacher sizes. For *Combined Augs*, the combination of these three augmentation strategies, the agreement drops even further, to just $60\%$ in self-distillation!

Our intuition about how knowledge distillation should work largely hinges on the assumption that after distillation the student matches the teacher on the distillation set. However, the results presented in this section suggest that in practice the optimization method is unable to achieve high fidelity *even on the distillation dataset* when extensive data augmentation or synthetic data is used. The inability to solve the optimization problem undermines distillation: in order to find a student that would match the teacher on all inputs, we need to at least be able to find a student that would match the teacher on all of the distillation data.

**Optimization and the train-test fidelity gap.** Notably, despite having the lowest train agreement, the *Combined Augs* policy results in better test agreement than other polices with better train agreement (Figure 3). This result highlights a fundamental trade-off in knowledge distillation: the student needs many teacher labels match the teacher on test, but introducing examples not in the teacher train data makes matching the teacher on the distillation data very difficult.

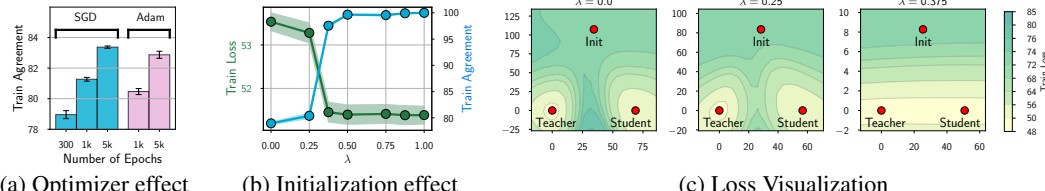

(a) Optimizer effect      (b) Initialization effect      (c) Loss Visualization

Figure 6: **Optimization and distillation**: self-distillation with ResNet-20s with LayerNorm on CIFAR-100. **(a)**: Final train agreement for SGD and Adam optimizers. Training longer improves agreement, but it remains below $85\%$ even after $5k$ epochs. **(b)**: Final train loss and agreement when the initialization is a convex combination of teacher and random weights, $\theta_s = \lambda\theta_t + (1 - \lambda)\theta_r$. **(c)**: Projections of the distillation loss surface on the plane intersecting $\theta_t$, the initial student weights, and the final student weights for different $\lambda$. When $\lambda$ is small, the student converges to a suboptimal solution with low agreement. The uncertainty regions correspond to $\mu \pm \sigma$, estimated over 3 trials.

## 6.2    Why is train agreement so low?

**A simplified distillation experiment.** To simplify our exploration, we focus on self-distillation of a ResNet-20 on CIFAR-100. We use the *Baseline* data augmentation strategy, as we found that a ResNet-20 student is unable to match the teacher on train even with basic augmentation. We also replace the BatchNorm layers [25] in ResNet-20 with LayerNorm [3], because we found that with BatchNorm layers even when the teacher and the student have identical weights, they can make different predictions due to differences in the activation statistics accumulated by the BatchNorm layers. Layer normalization does not collect any activation statistics, so the student will match the teacher as long as the weights coincide.

**Can we solve the optimization problem better?** We verify that the distillation fidelity cannot be significantly improved by training longer or with a different optimizer. By default, in our experiments we use stochastic gradient descent (SGD) with momentum, train the student for 300 epochs, and use a weight decay value of $10^{-4}$. In Figure 6 we report the results for the SGD and Adam [27] optimizers run for $1k$ and $5k$ epochs without weight decay. Switching from SGD to Adam only reduced fidelity.

For both optimizers, training for more epochs does slightly improve train agreement. In particular, with SGD we achieve $83.3\%$ agreement when training for $5k$ epochs compared to $78.95\%$ when training for 300 epochs. It is possible, though unlikely, that if we train for even more epochs the train agreement could reach $100\%$. However, training for $5k$ epochs is significantly longer than what is typically done in practice (100 to 500 epochs). Furthermore, the improvement from $1k$ to $5k$ epochs is only about $2\%$, suggesting that we would need to train for tens of thousands of epochs, even in the optimistic case that agreement improves linearly, in order to get close to $100\%$ train agreement.

**The distillation loss surface hypothesis**: If we cannot perfectly distill a ResNet-20 on CIFAR-100 with any of the interventions we have discussed so far, we now ask if there is any modification of the problem that *can* produce a high-fidelity student.

In the self-distillation setting, we do know of at least one set of weights that is optimal w.r.t. the distillation loss — the teacher's own weights $\theta_t$. Letting $\theta_r$ be a random weight initialization, in Figure 6 (a) we examine the effect of choosing the student initialization to be a convex combination of the teacher and random weights, $\theta_s = \lambda\theta_t + (1 - \lambda)\theta_r$. After being initialized in this way, the student was trained as before. In other words $\lambda = 0$ corresponds to a random initialization and $\lambda = 1$ corresponds to initializing the student weights at the final teacher weights.

We find that if the student is initialized far from the teacher ($\lambda \leq 0.25$), the optimizer converges to a sub-optimal value of the distillation loss, producing a student that significantly disagrees with the teacher. However at $\lambda = 0.375$ there is a sudden change. The final train loss drops to the optimal value and the agreement drastically increases, and the behavior continues for $\lambda > 0.375$. To further investigate, in Figure 6 (c) we visualize the distillation loss surface for $\lambda \in \{0, 0.25, 0.375\}$ projected on the 2D subspace intersecting $\theta_t$, the initial student weights, and the final student weights. If the student is initialized far from the teacher ($\lambda \in \{0, 0.25\}$), it converges to a distinct, sub-optimal basin of the loss surface. On the other hand, when initialized close to the teacher ($\lambda = 0.375$), the student converges to the same basin as the teacher, achieving nearly 100% agreement.

| Init. | Agree. (↑) | KL (↓) | CKA (↑) | | |
|---|---|---|---|---|---|
| | | | Stage 1 | Stage 2 | Stage 3 |
| Rand. | 77.174 (0.352) | 0.836 (0.016) | 0.939 (0.017) | 0.925 (0.027) | 0.885 (0.011) |
| Teach. | 77.098 (0.238) | 0.838 (0.020) | 0.951 (0.017) | 0.937 (0.020) | 0.890 (0.015) |

Table 1: We examine whether fidelity can be improved in the context of ResNet-20 self-distillation on CIFAR-100 if the teacher and student share the same weight initialization. All metrics are computed on the test set. A shared initialization does make the student slightly more similar to the teacher in activation space (measured by CKA), but in function space the results are indistinguishable from randomly initialized students. We report the mean and standard deviation, estimated from 10 trials. The average teacher accuracy was 70.522 (0.412).

**Is using the initial teacher weights enough for good fidelity?** If good fidelity can be obtained by initializing the student near the *final* teacher weights, it is possible that similar results could be obtained by initializing the student at the *initial* teacher weights. In Table 1 we compare students distilled from random initializations with those initialized at the initial teacher weights. In addition to the metrics reported in the rest of the paper, we also include the centered kernel alignment (CKA) [28] of the preactivations of each of the teacher and student networks. There is a small increase in CKA, indicating that sharing an initialization between teacher and student does increase alignment in activation space, but functionally the students are identical to their randomly initialized counterparts – there is no observable change in accuracy, agreement, or predictive KL when compared to random initialization.

To summarize, we have at last identified a root cause of the ineffectiveness of all our previous interventions on the knowledge distillation procedure. Knowledge distillation is unable to converge to optimal student parameters, even when we know a solution and give the initialization a small head start in the direction of an optimum. Indeed, while identifiability can be an issue, in order to match the teacher on all inputs, the student has to at least match the teacher on the data used for distillation, and achieve a near-optimal value of the distillation loss. Furthermore, the suboptimal convergence of knowledge distillation appears to be a consequence of the optimization dynamics specifically, and not simply initialization bias. In practice, optimization converges to sub-optimal solutions, leading to poor distillation fidelity.

## 7  Discussion

Our work provides several new key findings about knowledge distillation:

- *Good student accuracy does not imply good distillation fidelity:* even outside of self-distillation, the models with the best generalization do not always achieve the best fidelity.

- *Student fidelity is correlated with calibration when distilling ensembles:* although the highest-fidelity student is not always the most accurate, it is always the best calibrated.

- *Optimization is challenging in knowledge distillation:* even in cases when the student has sufficient capacity to match the teacher on the distillation data, it is unable to do so.

- *There is a trade-off between optimization complexity and distillation data quality:* Enlarging the distillation dataset beyond the teacher training data makes it easier for the student to identify the correct solution, but also makes an already difficult optimization problem harder.

In standard deep learning, we are saved by not needing to solve the optimization problem well: while it true that our training loss is highly multimodal, properties such as the flatness of good solutions, the inductive biases of the network, and the implicit biases of SGD, often enable good generalization in practice. In knowledge distillation, however, good fidelity is directly aligned with solving what turns out to be an exceptionally difficult optimization problem.

## Acknowledgements

The authors would like to thank Gregory Benton, Marc Finzi, Sanae Lotfi, Nate Gruver, and Ben Poole for helpful feedback. This research is supported by an Amazon Research Award, NSF I-DISRE 193471, NIH R01DA048764-01A1, NSF IIS-1910266, and NSF 1922658NRT-HDR: FUTURE Foundations, Translation, and Responsibility for Data Science. Samuel Stanton is also supported by a United States Department of Defense NDSEG fellowship.

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
