# Does Knowledge Distillation Really Work?
# Supplementary Material

**Supplement Outline**:

    A. Implementation details for all experiments.

    B. Additional experiments with the teacher ensemble size ablation.

    C. Experiments addressing spurious explanations for poor student fidelity.

    D. NeurIPS checklist responses.

## A  Implementation details

Here we briefly describe key implementation details to reproduce our experiments. Data augmentation details are given in A.1, followed by architecture details in A.2, and finally training details are provided in A.3. The reader is encouraged to consult the included code for closer inspection.

### A.1  Data augmentation procedures

Some of the data augmentation procedures we consider attempt to generate data that is close to the train data distribution (standard augmentations, GAN, mixup). Others (random noise, out-of-domain data) produce data for distillation that the teacher would never encounter during normal supervised training. In particular, we compare the following augmentation procedures:

**Baseline augmentations**    As a baseline, we use the same data augmentation strategy that was used to train the teachers during distillation: we apply random horizontal flips ($p = 0.5$) and random shifts via pad and random-crop with a 4 pixel pad width. In all of the configurations we consider in this section we use this set of augmentations along with other strategies, unless stated otherwise.

**Conventional image transformations**    Standard data augmentations used in computer vision [10]: random rotations by up to 20 degrees, random vertical flips, color jitter and all possible combinations.

**Mixup**    Mixup is an effective regularization technique originally proposed to increase generalization and robustness of deep networks [19, 17]. Instead of training on original dataset, the network is trained on convex combination of images with targets mixed in the same way. We adapt mixup to knowledge distillation as follows: on each iteration we construct random pairs of inputs $\mathbf{x}, \mathbf{x}'$ from the training set and mix them as $\lambda \cdot \mathbf{x} + (1 - \lambda) \cdot \mathbf{x}'$, where the coefficient $\lambda$ is sampled uniformly on $[0, 1]$[1].

**Synthetic GAN-generated images**    We use a Spectral Normalization GAN (SN-GAN) trained on CIFAR-100 [11] to generate synthetic data for distillation. We used the same pretrained SN-GAN (FID $= 74.2617$, IS $= 6.6023$) for all experiments. Our synthetic augmentation procedure was the following: for each minibatch of real training data, we concatenated synthetic images sampled from a pretrained SN-GAN at a ratio of 1 synthetic image to 4 real images.

---

[1]Note that unlike in the original mixup procedure we are only mixing the inputs and we use the predictions of the teacher on the mixed inputs as the target for the student.

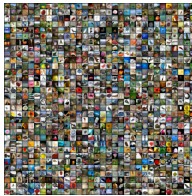 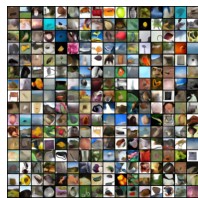 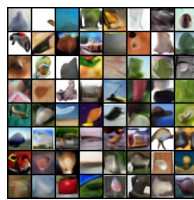 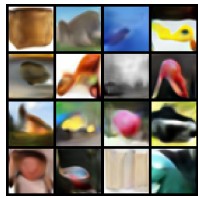

Figure 7: Sample images from our SN-GAN (FID = 74.2617, IS = 6.6023) trained on CIFAR-100.

**Random noise**    To observe the effect of unnatural images in the distillation dataset we augment with images sampled pixel-wise from uniform $[0, 1]^d$. During distillation each image in a minibatch is randomly resampled with probability $p = 0.2$.

**Out-of-domain data**    Finally, we consider using images from the SVHN dataset [13] which is semantically unrelated to the target CIFAR-100 dataset.

We use the `torchvision.transoforms` package [14] to implement the augmentations from the Baseline Augmentations and Conventional Image Transformations categories:

- Horizontal flips: `torchvision.transforms.RandomHorizontalFlip()`
- Random shifts: `torchvision.transforms.RandomCrop(size=<input_size>, padding=4)`
- Vertical flips: `torchvision.transforms.RandomVerticalFlip()`
- Color jitter: `torchvision.transforms.ColorJitter(brightness=0.2, contrast=0.2, saturation=0.2, hue=0.2)`
- Random rotations: `torchvision.transforms.RandomRotation(degrees=20)`

## A.2 Network architectures

**Image classifiers**    For experiments on CIFAR-100 we used preactivation ResNets with batchnorm, skip connections [5], and the standard three-stage macro-structure, varying the number of layers in each stage (i.e. the depth of the network). For all choices of depth we used the same number of filters in each stage (16, 32, and 64, respectively). In Section C.2 we use a VGG-16 network without batch-normalization, with implementation directly adapted from `https://github.com/pytorch/vision/blob/master/torchvision/models/vgg.py`. For experiments on MNIST and EMNIST we used a 5-layer LeNet [7]. For ImageNet we used ResNet50 networks for both teacher and students.

**Image generators**    We used the standard three-stage ResNet architectures for the SN-GAN generator and discriminator from Miyato et al. [11]. The generator latent dimension was 128, and each generator stage had 256 filters. The discriminator had 128 filters in each stage. Sample images are shown in Figure 7.

**Text classifier**    In section C.4 on IMDB sentimental analysis problem, we use a bidirectional LSTM recurrent neural network with 2 layers, embedding size 128 and LSTM cell size 70.

## A.3 Training procedures

In our experiments, we consider pre-activation ResNet networks with depths 20, 56 and 110 [5]. We evaluate on MNIST/EMNIST [1] and CIFAR-100 [6], focusing primarily on the latter. We chose CIFAR-100 rather than CIFAR-10 because increasing the problem difficulty increases the gap in performance between a single model and an ensemble, making significant trends more apparent. We independently train each network in the teacher-ensemble to minimize $\mathcal{L}_{\mathrm{NLL}}$ for 200 epochs, and we distill each student by training it to minimize $\mathcal{L}_{\mathrm{KD}}$ for 300 epochs (i.e. we take $\alpha$ in $\mathcal{L}_s$ to be 0). Note that in the literature one typically sees $\alpha > 0$. We have chosen $\alpha = 0$ so that our objective reflects our aim of producing the highest fidelity student possible. We use an SGD optimizer with

initial learning rate $5 \times 10^{-2}$ and cosine annealing learning rate schedule. We produce augmented datasets for distillation by sampling images from a set of specified sources, including rotations or color jitter applied to ground truth images, uniform 'white noise' images, or synthetic GAN-generated images. Unless specified otherwise, the only augmentations applied when training the teacher were the standard random horizontal flip ($p = 0.5$) and padded random crop (4 pixel pad width), regardless of the choice of distillation dataset.

**Teacher image classifiers:** The teacher models were trained through the standard empirical cross-entropy loss for 200 epochs with a batch size of 256 using SGD with momentum (0.9 momentum weight) and weight decay of $1.0 \times 10^{-4}$. We used a cosine annealing learning rate schedule with $\eta_{\max} = 0.1$, $\eta_{\min} = 0$. For data augmentation we used random horizontal flips ($p = 0.5$) and random crops (padding width 4).

**Student image classifiers:** Our student models were distilled through the temperature-scaled teacher-student cross-entropy with varying temperatures $\tau$ for 300 epochs, with a batch size of 128 using SGD with Nesterov momentum (0.9 momentum weight) and weight decay of $1.0 \times 10^{-4}$. We used a cosine annealing learning rate schedule with $\eta_{\max} = 5.0 \times 10^{-2}$, $\eta_{\min} = 1.0 \times 10^{-6}$. For details on the data augmentation procudures we considered, the reader is directed to Appendix A.1.

**Image generators:** For synthetic image generation we trained SN-GAN models with the hinge discriminator loss from Miyato et al. [11]. We trained the generator for 100K gradient steps with a batch size of 128. For each generator step, we took 5 discriminator steps. We used Adam ($\beta_1 = 0$, $\beta_2 = 0.9$) and a linearly decayed learning rate $\eta_{\max} = 2.0 \times 10^{-4}$, $\eta_{\min} = 1.0 \times 10^{-6}$. We used random horizontal flips ($p = 0.5$) as data augmentation for the discriminator. To evaluate FID and IS scores, we used 5K samples from the generator and the pretrained PyTorch Inception-v3 networks[2]. For the discriminator and generator architectures the reader is referred to Appendix A.2.

**Text classifiers:** For text classification on IMDB dataset, we train LSTM networks for 100 epochs with learning rate $10^{-2}$, weight decay $10^{-3}$ and batch size 100 sequences. For the data loader, we use 100 as maximum sequence length and filter out tokens in the vocabulary that are present less than 10 times.

**ImageNet experiments:** we trained the teachers with weight decay $10^{-4}$ and did not use weight decay for training the students. We trained both the teachers and the students for 90 epochs using the SGD optimizer with momentum 0.9 and a cosine decay learning rate schedule with a linear learning rate ramp-up for 5 epochs to the initial value of 0.1. We used a batch size of 1024.

---

[2]`https://pytorch.org/docs/stable/torchvision/models.html?highlight=inception#torchvision.models.inception_v3`

# B Additional understanding experiments

In this section we include additional experimental results that were not included in the main text in the interest of clarity, but are still noteworthy to those seeking a deep understanding of the behavior of knowledge distillation.

## B.1 Understanding the effect of teacher capacity on the distillation labels

In this subsection, we explore the qualitative effect of teacher ensemble size, network depth, and distillation temperature on the predictive distributions on train and test, to get a better understanding of what the students are being asked to emulate.

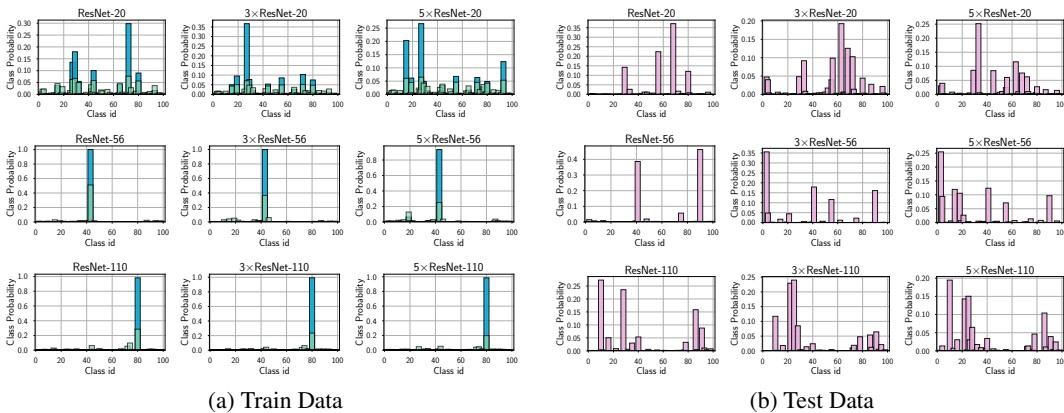

(a) Train Data        (b) Test Data

Figure 8: Teacher predictive distributions for example images from CIFAR-100 train (**left**) and test (**right**). For train examples we show the distribution when $\tau = 1$ in blue and the tempered distribution when $\tau = 4$ in green. For test examples we only show $\tau = 1$. Each row corresponds to a different teacher depth, and the column corresponds to the number of ensemble components.

Since deep ensemble components are typically large networks that achieve almost 100% accuracy on train with very high confidence, it is tempting to assume that each ensemble component conveys effectively the same information when used for distillation. One consequence of that assumption would be that adding ensemble components would produce little or no improvement in the student if the distillation was performed on train. In fact, we find that although the component networks are indeed very confident on the train, there is sufficient variation in their predictive distributions for the student to benefit significantly. In Figure 8 we provide examples of teacher logits, varying the ensemble size, network depth, and temperature, comparing examples from CIFAR-100 train and test.

As discussed in Section 3.1 in order to benefit from soft teacher labels, one must choose $\tau$ large enough that the student directs some capacity towards mimicking the smaller teacher logits (Figure 4, bottom). If $\tau$ is chosen too small (e.g. $\tau = 1$), then the student distilled from a 3-component ResNet56 ensemble is no better than a student distilled from a single network. The improvements in student performance and fidelity taper off fairly quickly as ensemble components are added.

The correct choice of $\tau$ depends on the level of confidence the teacher has on train. ResNet56 networks achieve nearly 100% accuracy on train with high confidence, so a temperature like $\tau = 4$ works well. When ResNet20 networks are used (networks which are not capable of perfectly fitting CIFAR-100), we see that lower temperatures can be used, although $\tau = 4$ still outperforms other choices (Figure 9). The reason lower temperatures work with ResNet20 teachers on CIFAR-100 is because ResNet20 networks do not attain 100% accuracy on train, so the teacher logits are much less sharply peaked (see also Figure 8, top row).

## B.2 Understanding the effect of teacher ensemble size on distillation

In Figure 10 we demonstrate the effect of increasing the number ($m$) of teacher ensemble ResNet56 components on test accuracy and agreement. In the main text we only considered teacher ensembles

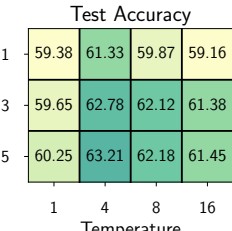
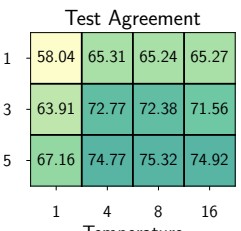

Figure 9: Subsampled CIFAR-100 experiment performed with ResNet20 networks. ResNet20 networks are much less confident on train than ResNet56 networks. As a result increasing the ensemble size will improve the student even with a small temperature setting $\tau = 1$.

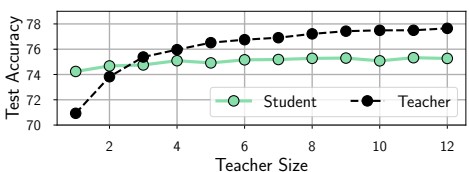
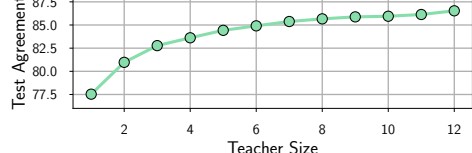

Figure 10: The effect on accuracy (**left**) and agreement (**right**) of the number of models ($m$) in the teacher ensemble ($\alpha = 0, \ \tau = 1$). Student accuracy quickly saturates as $m$ increases, despite continuing improvements in teacher accuracy. The teacher-student agreement continues to improve after the accuracy has saturated.

with up to 5 components – here we provide results for up to 12 components. Although it is plausible that ensembles with more components would have more complex predictive distributions that would be difficult for a single student to match, in reality we see the exact opposite. Deep ensembles with *more* components are easier to emulate (indicated by higher agreement). One possible explanation is that adding more ensemble components smooths the logits of unlikely classes, making the distribution easier to match. Closer investigation into this phenomenon could potentially yield insights into how to improve distillation fidelity in general.

In agreement with the results on self-distillation [3, 12], we see that the student is more accurate than the teacher when $m = 1$. However the accuracy of the student does not substantially improve as we increase the number $m$ of models in the teacher ensemble past 4, even though both the accuracy of the teacher and the teacher-student agreement continue to increase monotonically with $m$.

### B.3   Detailed results for distillation with heavy data augmentation

In Figure 11 we report more detailed results for the experiment in Figure 3 (in the main text). In particular, for the sake of simplicity we only reported results for $m = 5$ in the main text. Here we report results for $m = 1$ and $m = 3$ as well for comparison.

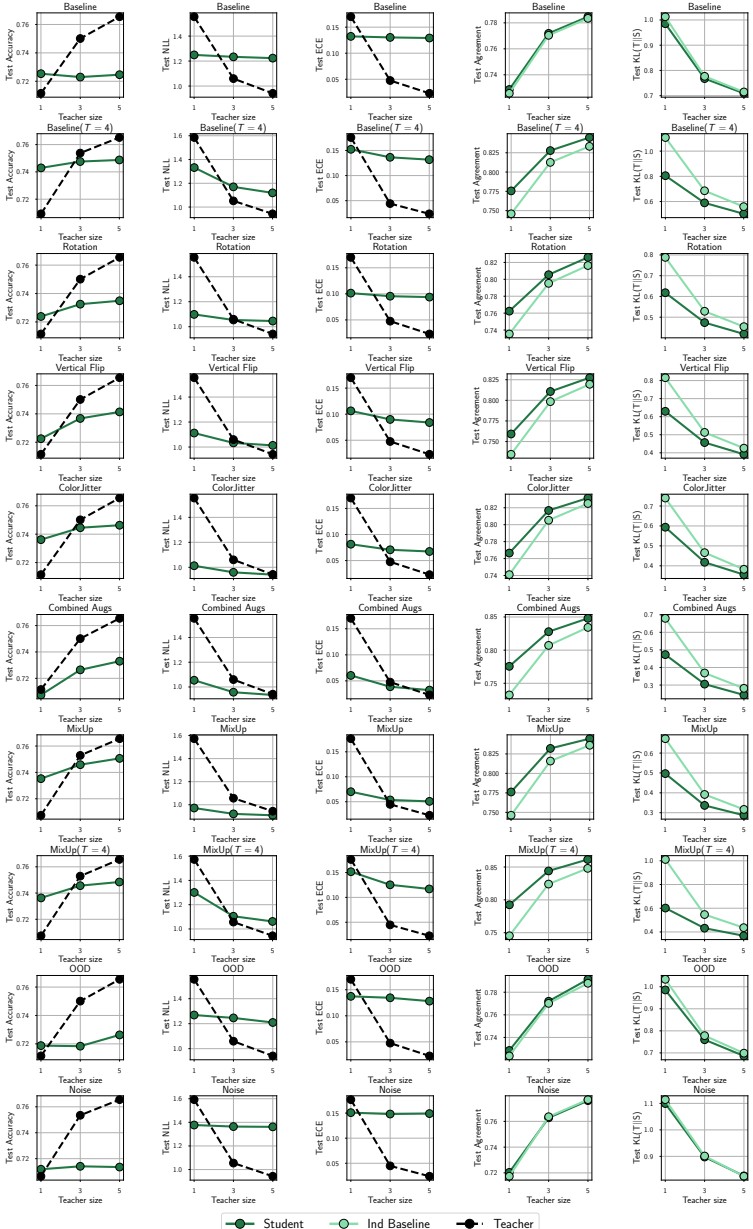

Figure 11: Detailed results for the experiment in Section 5.1. Each row corresponds to a different augmentation procedure, and each column is a different evaluation metric. Notably, we see that the student distilled with mixup and $\tau = 1$ is the best overall in terms of NLL (though not test accuracy) beating even the teacher-ensemble for all values of $m$. The independent baseline serves as a reference to aid in the interpretation of fidelity metrics.

# C   Addressing alternative causes of poor fidelity

In this section we provide evidence contrary to other possible explanations of low student fidelity posited in Section 4.4. In section C.1 we demonstrate that increasing student capacity does not substantially improve fidelity. We also show that poor distillation fidelity is not specific to neural network architecture, dataset scale and data domain, and is observed for VGG networks (section C.2), larger-scale ImageNet dataset (section C.3) and IMDB sentiment analysis classification with LSTM networks (section C.4). Further, in section C.5 we demonstrate that the common practice of showing the student both the real labels (when available) and the teacher labels tends to decrease fidelity.

## C.1   Capacity: is the student capable of emulating the teacher?

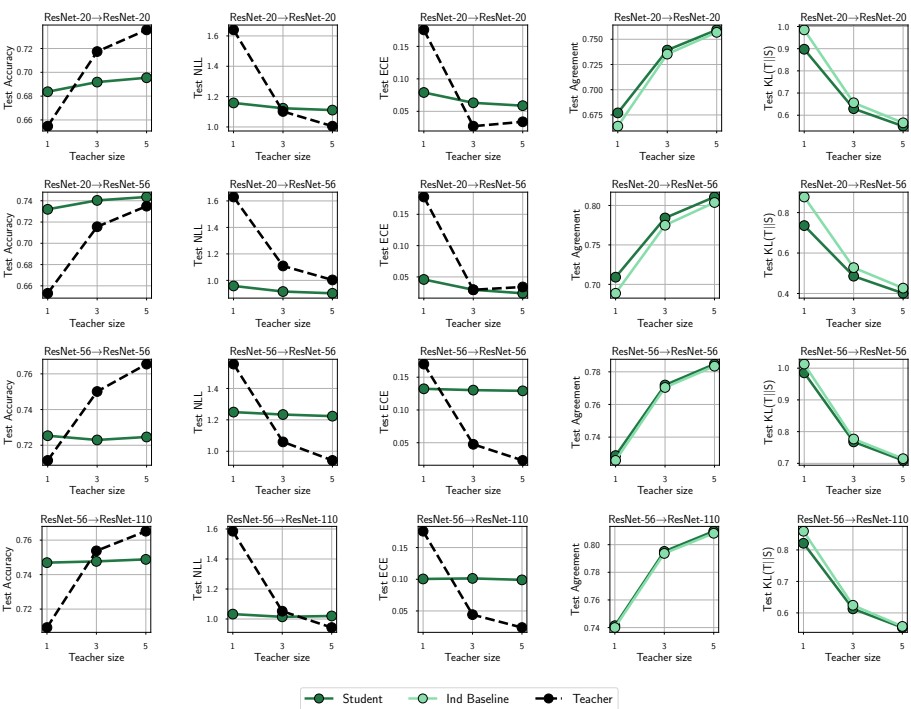

Figure 12: Here we show the effect of increasing the student capacity, holding the teacher capacity fixed. The top two rows correspond to ResNet20 teacher-ensemble components with ResNet20 and ResNet56 students, respectively. The bottom two rows are similarly ResNet56 teacher components with ResNet56 and ResNet110 students. The column corresponds to the evaluation metric. Increasing student capacity from 20 to 56 provides some benefit to both accuracy and fidelity, but increasing student capacity from 56 to 110 improves only accuracy.

One possible cause of low student fidelity when distilling ResNet ensembles on CIFAR-100 is that a single student network does not have *capacity* to perfectly emulate an ensemble of multiple networks. This explanation is already rendered unlikely by our similar observations in the context of self-distillation. Nevertheless in Figure 12 we demonstrate the effect of increasing student capacity beyond that of the individual teacher components as additional contrary evidence to the capacity explanation. Increasing the student capacity does slightly improve fidelity – doubling the student network depth results in a 2% to 3% improvement in test agreement. If capacity were a primary cause of low fidelity, we would expect a much larger effect on distillation fidelity when the student capacity is significantly increased.

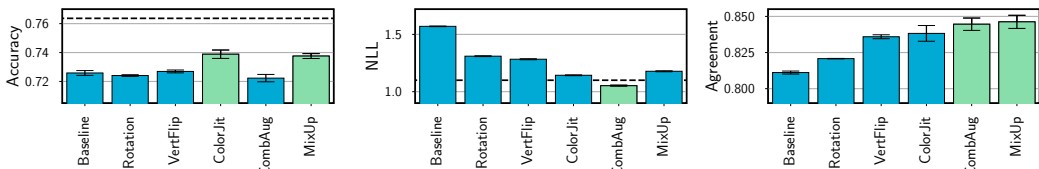

Figure 13: Test accuracy, negative log-likelihood and teacher-student agreement when distilling a 5-component VGG-16 teacher ensemble into a VGG-16 student on CIFAR-100 with varying augmentation policies. The best performing policy is shown in green, results averaged over 3 runs and error bars indicate $\pm\sigma$. The results are generally analogous to the results for the ResNet-56 architecture reported in Section 5.1. *MixUp* and *ColorJit* provide the best student accuracy, while *CombAug* provides the best NLL. *CombAug* and *MixUp* provide the best teacher-student agreement.

## C.2 Architecture: is low fidelity an artifact of using ResNets?

Another possible explanation of low student fidelity in our experiments is our choice of network architecture. ResNet-style backbones are ubiquitous across most computer vision tasks, so even were the issue restricted to ResNets it would merit close investigation. Nevertheless, in the interest of empirical rigor we repeat the augmentation ablation in Section 5.1 with VGG networks and a subset of the augmentation policies. In Figure 13, the teacher is a 5-component ensemble of VGG-16 networks trained with the *Baseline* augmentation policy (horizontal flips and random crops). We report the student accuracy, negative log-likelihood and teacher-student agreement for a VGG-16 student trained with different data augmentation policies.

The results are generally analogous to the ones for ResNet-56 presented in Section 5.1. The *CombAug* augmentation strategy underperforms all other strategies, including *Baseline*, on student accuracy, but provides the best results on NLL and only slightly loses to *MixUp* on teacher-student agreement. This result again highlights that the best augmentation policies for generalization do not necessarily provide the best distillation fidelity. Finally, regardless of the augmentation strategy, the agreement on test does not exceed $85\%$.

## C.3 Dataset: does increasing the scale of the dataset increase fidelity?

We provide the results for distilling ensembles of $1$, $3$ and $5$ ResNet-50 teachers into a single ResNet-50 model in Table 1. For each setting, we report the results averaged over 3 independent runs. The results further validate our CIFAR-100 experiments. In particular, top-1 agreement is again in the $80 - 90\%$ range, adding more ensemble components to the teacher improves student accuracy and fidelity, and both the accuracy and fidelity gap between teacher and student can be observed.

| Dataset | Teach. Size | Teach. Acc. ($\uparrow$) | Stud. Acc. ($\uparrow$) | Agree. ($\uparrow$) | KL ($\downarrow$) |
|---------|-------------|--------------------------|-------------------------|---------------------|-------------------|
| IMDB | 1 | 79.361 (0.132) | 80.353 (0.198) | 86.488 (0.521) | 0.124 (0.012) |
| | 3 | 81.807 (0.129) | 81.129 (0.057) | 89.832 (0.349) | 0.064 (0.003) |
| | 5 | **82.216 (0.207)** | **81.167 (0.196)** | **90.793 (0.180)** | **0.052 (0.001)** |
| ImageNet | 1 | 0.748 (0.001) | 0.753 (0.001) | 0.855 (0.001) | 0.217 (0.002) |
| | 3 | 0.764 (0.001) | 0.755 (0.001) | 0.878 (0.001) | 0.157 (0.001) |
| | 5 | **0.767 (0.001)** | **0.756 (0.001)** | **0.884 (0.001)** | **0.142 (0.001)** |

Table 1: Distillation results when the dataset is varied. All metrics are computed on the test set. We used bidirectional LSTM networks for IMDB and ResNet-56 networks for CIFAR-100 and ImageNet. Across all datasets we see the following consistent behavior: 1) larger teacher ensembles are more accurate and easier to distill, and 2) teacher-student disagree on at least 10% of test points.

## C.4 Data domain: is low fidelity specific to image classification?

To expand out results beyond the image domain, we also demonstrate the knowledge distillation results when distilling LSTM text classifiers on IMDB sentiment analysis data in Table 1. We distill ensembles of 2-layer bidirectional LSTM teachers into a single bidirectional LSTM of the

same architecture. Note that like in our other results, the teacher accuracy is strictly improving, whereas the student accuracy ceases to improve from 3 to 5 teacher ensemble components, which is associated with a similar lack of improvement in agreement.

### C.5 Does showing the student ground truth labels improve fidelity?

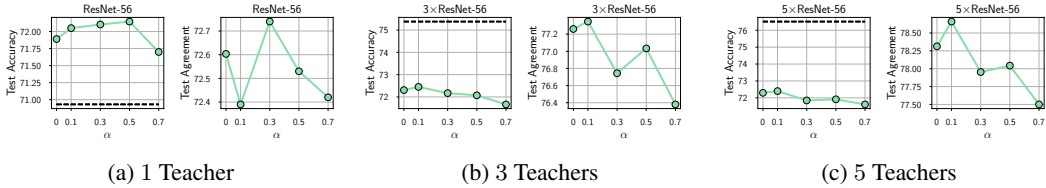

(a) 1 Teacher          (b) 3 Teachers          (c) 5 Teachers

Figure 14: Results ablating $\alpha$ ($\tau = 1$). Taking $\alpha > 0$ can improve student accuracy in the self-distillation regime, but does not consistently improve teacher-student agreement. When $k > 0$ there is a slight benefit at $\alpha = 0.1$, after which the effect is negative for both accuracy and agreement.

In Figure 14 we investigate the effect of the relative weight of the distillation loss terms $\mathcal{L}_{\mathrm{NLL}}$ and $\mathcal{L}_{\mathrm{KD}}$ when distilling teacher-ensembles with ResNet56 components into a ResNet56 student on CIFAR-100 with $\tau = 1$. We observe that in the self-distillation regime taking $\alpha > 0$ improves test accuracy, but not test agreement. When $k > 0$, there is a slight benefit when $\alpha = 0.1$, but for most values tried the effect was deleterious to both accuracy and fidelity.

## D  Checklist responses

### D.1  Limitations

Our work thoroughly investigates the topic of knowledge distillation fidelity in the context of image classification, where distillation is regularly applied, and leaves other domains like language modeling for future work. Our work also does not consider variants of knowledge distillation that incorporate information from the weights or intermediate activations of the teacher, which are sometimes used to provide additional signal to students [16], since such variants tend to be very specific to the teacher-student architecture pair and are not particularly well-suited to ensemble distillation.

### D.2  Social impact

As awareness of the need to mitigate the resource consumption of large machine learning models grows, distillation becomes an increasingly relevant research topic. Although distillation does not reduce the initial cost of training a large model, it allows smaller, more efficient models to emulate the behavior of the large model at a fraction of the test-time resource consumption. Furthermore, knowledge distillation holds promise for transferring powerful representations from large black-box models to smaller, more interpretable models, which could greatly increase the trustworthiness of machine learning models for high-stakes decision-making. Our findings highlight that we cannot assume that a student provides a faithful representation of a teacher — a teacher which we may trust — simply because the student is generalizing well on a set of tasks. We hope that our work will inspire a line of research trying to *understand* knowledge distillation, particularly outside of the self-distillation setting.

### D.3  Estimated compute used

This paper required approximately 1k GPU hours, or about 40 GPU days, by our Fermi estimate,

10 GPU hrs/seed $\times$ 1 seed per variant $\times$ 10 variants per experiment $\times$ 10 experiments $= 1,000$ hrs.

Our experiments were conducted on a range of NVIDIA GPUs, including K80s, RTX 2080 Tis, and Titan RTXs. The ImageNet experiments were conducted on 16 TPU-V3 devices.

### D.4 Assets

**Datasets**:

- MNIST, LeCun et al. [8], Creative Commons Attribution-Share Alike 3.0 License
- EMNIST, Cohen et al. [1]
- CIFAR-100, Krizhevsky et al. [6]
- ImageNet, Deng et al. [2],
- IMDB, Maas et al. [9].

**Software**:

- Python 3, Van Rossum and Drake [18], PSF License Agreement,
- NumPy, Harris et al. [4], BSD License,
- PyTorch, Paszke et al. [15], BSD License.