# OpenReview forum: "Does Knowledge Distillation Really Work?"
_NeurIPS.cc/2021/Conference — NeurIPS 2021 Poster_

### Official Review · Reviewer_E993 · 2021-06-30

**Rating:** 6
**Confidence:** 4

**Summary:**

This paper uses a set of experiments to investigate whether/how knowledge distillation works. Two measurements are used: (1) fidelity to measure how well the student's output matches the teacher's, and (2) generalization, i.e., the top-1 accuracy. The authors claim that (1) low fidelity can be mitigated by augmenting the distillation set but the generalization is hardly improved by doing so, and (2) the difficulty in optimization is the key reason why the student cannot mimic the teacher well. Experiments with ResNet on CIFAR-100 support the above claims.

**Limitations And Societal Impact:**

The authors addressed the limitations in Appendix.

**Main Review:**

First of all, understanding why/how KD works is a valuable study. Most previous KD approaches are built upon Hinton's paper [1] by matching more elements between the teacher and the student, rarely explaining why KD works in that way. From this point of view, I acknowledge the motivation of this paper.

My main concerns are as follows:
 - Although the observations proposed by the paper is interesting, it, unfortunately, does not result in constructive suggestions on how to improve the training of the student model via KD. For the first observation (Section 5), the authors find that data augmentation can slightly improve fidelity, but the improvement over different kinds of augmentation strategies seems random, i.e., it is observed that MixUp is a better strategy but it is unclear why it works better than others. Moreover, the results in Figs. 3 and 4 show that there is no obvious relation between fidelity and generalization (accuracy), i.e., as claimed at the end of Section 5, 'the evidence does not support blaming poor distillation fidelity on the wrong choice of distillation data'.  For the second observation (Section 6), the paper shows some evidence that it is difficult for the student to match the teacher on the distillation data. Similarly, there is still no solution on how to solve this issue. The conclusion made at the end of Section 6 'the student has to as least match the teacher on the data used for distillation, and achieve a near-optimal value of the distillation loss' is kind of weak and high-level, which is difficult to implement explicitly in practice. With these observations, it seems that we still have to do KD like how we did before.

 - The observations found by this study are intuitive but not surprising (at least not 'surprising enough'), especially for the second observation. It is observed that more distillation data lowers the training agreement but improves the test agreement. This phenomenon is quite similar to what we observe in terms of training and test accuracy, i.e., data augmentation lowers the training accuracy but improves the test accuracy.

 - The observations and claims are only supported by the CIFAR-100 and ResNet experiments (although there are a few experiments with VGG in Appendix), which weakens the findings in the paper. It would be much better if the authors could prove that these findings still hold (1) with different datasets, especially on large-scale datasets like ImageNet, (2) when the teacher and the student are with different network structures, and (3) with more KD approaches besides [1].

[1] Hinton, Geoffrey, Oriol Vinyals, and Jeff Dean. "Distilling the knowledge in a neural network." arXiv preprint arXiv:1503.02531 (2015).


Minor issues:

1. Is there a way that can quantitatively measure the similarity between two data distributions? If there is not, it is difficult to estimate the distribution gaps between the original training samples and the noise, OOD data, and pseudo samples generated with GAN.

2. In Fig. 5(left), why is the training agreement of 5 teachers higher than that of 3 teachers?

**Time Spent Reviewing:**

12

---

> ### Author Response · Authors · 2021-08-10
> **Individual Author Response: E993**
>
> Thank you for your time and feedback! We share your desire for practical, constructive insights, and we believe there are many to be gained from our work. For your convenience we have enumerated these suggestions in the separate post with our general response, which also includes new ImageNet and IMDB experiments inspired by your comments. We hope you can consider our response in your final assessment.
>
> 1. Novelty of the results: some of our findings, such as the effect of data augmentation on train accuracy, might be expected, but are still valuable to confirm. Other experiments did not conform to our expectations. In particular, we expected data recycling to be worse than a fresh draw from the same distribution, but it was not (Section 5.2). Similarly the finding that larger ensembles are easier to distill than small ones is at once a practical suggestion and a counterintuitive finding (as your related question demonstrates), since one might think that larger ensembles could potentially have more complex decision boundaries that would be hard for a single network to emulate. Please also see our separate general post for a further discussion of novelty.
>
> 2. Scope of the results: there is always a tension between limiting experimental confounders and broadening the scope of the results. Inspired by your comments, we are happy to provide new results for ImageNet and IMDB (text classification) as additional evidence for our findings, which we present in the general post. Varying the architecture between teacher and student, though common in practice, is a serious confounder because the capacity and inductive biases of the student and teacher are no longer the same. We chose to focus on ‘vanilla’ KD because it is the most natural choice when distilling ensembles, which was the original motivation for the project. More sophisticated distillation methods are usually designed for a single teacher model and come laden with additional design choices that make analysis all the more difficult.
>
> 3. Estimating the distance between distributions: metrics such as MMD could be used to estimate the distance between distributions, but finding a suitable kernel is non-trivial.
>
> 4. Why are larger ensembles easier to distill? In general, we found that larger ensembles were easier to distill than smaller ones. One reason is that larger ensembles have smoother predictive distributions that are easier to fit (see Figure 12 in the appendix).

---

> > ### Comment · Reviewer_E993 · 2021-08-27
> > **post-rebuttal feedback**
> >
> > I'd like to thank the authors for the very detailed response. The newly added experiments in the general author response are very useful. Although I still think that this study did not provide a perfect solution to deal with the phenomena found in the paper, it is definitely a valuable work that inspires people to think about the rationale behind KD. This is also confirmed by other reviewers. Therefore, I increase my rating to 6: Marginally above the acceptance threshold.

---

> > > ### Author Response · Authors · 2021-08-27
> > > **Follow-up response to reviewer E993**
> > >
> > > Thanks for your feedback and support! We agree our work indicates that there are intruiging open questions around the topic of distillation, which we hope will motivate continued research on the subject.

---

### Official Review · Reviewer_CMC1 · 2021-07-15

**Rating:** 7
**Confidence:** 4

**Summary:**

This paper provides a novel view for understanding the knowledge distillation, the fidelity. By conducting rich experiments and analysis about the fidelity learning in various KD settings, the conventional insights about the 'mimic' operation in KD, the alignment of output distribution, should be with  more and deeper discussion.

**Limitations And Societal Impact:**

There are some questions about this paper:

1. Line66-71. It seems that this approximation only works in the situation that teacher netwrok makes the correct prediction. Please verify it.

2. Line 146 and line 159 are conflict to each other. I suppose that higher fidelity are not correlated with better generalizability.

3. Sec.5.1. Usually, the data augmentation is introduced by changing the desired invariant information in the data. For example, the 'crop' operation is used to ensure the model is scale-invariant in classification task. It is expected that there is little difference between the original data and its corresponding augmented ones. Therefore, I just wonder that if this part is reasonable. Since Sec.5.2 already gives good analysis and explanation of the identifiability probelm.

4. Line 211-213. Is this because the hard examples are indeed hard to learn for all networks? Thus, not the problem for knowledge distillation.

5. Line 325-330. Is this because the optimization task here is a non-convex problem, thus finding the optimal solution is impossible in most cases.

6. Line 335. What is the meaning of the 'calibration' here? Please explain the term in the paper.

**Main Review:**

This paper is novel, high quality, clear and significant.

**Time Spent Reviewing:**

4.5

---

> ### Author Response · Authors · 2021-08-10
> **Individual Author Response: CMC1**
>
> Thank you for your supportive review! We provide responses to your questions below. Please also see our separate general post, where we provide several new results, including ImageNet and IMDB.
>
> 1. On the approximation of the KD gradient: this approximation was introduced in the original KD paper [1], where it is explained in more detail. In general, since the KD loss only depends on the teacher and student labels, whether the teacher makes the ‘right’ prediction has no impact on the KD loss whatsoever.
>
> 2. Fidelity and generalization: It is important not to extrapolate too heavily from the self-distillation setting, which is only one of many ways in which knowledge distillation is used. Understanding the relationship between fidelity and generalization requires care. Fidelity and generalization are correlated, but not perfectly. Very accurate teachers and students will necessarily agree on many points, otherwise they could not both be accurate (see Section 3.2 for a simple example). When the teacher is better than the student, because it is a large model, or an ensemble, then good accuracy is highly aligned with good fidelity.
>
> 3. Data augmentation: It is important to note that the teachers were only trained with the baseline augmentation policy. Hence they are not even approximately invariant to the heavier data augmentation policies, such as CombinedAugs or MixUp.
>
> 4. Hard Examples? The results in Figure 5 are inconsistent with the issue being simply hard examples. The examples in D_0 are no easier or harder, on average, than the examples in D_1, and yet we see significant differences in the fidelity and accuracy that correspond to our other experimental findings.
>
> 5. Non-convexity: we agree whole-heartedly that non-convexity is a major issue here. Because of the success of SGD for objectives like the hard-label cross-entropy, one might be tempted to think we can solve superficially similar problems like distillation equally well, but as we show that is interestingly not the case for knowledge distillation.
>
> 6. Calibration: Calibration measures whether a model “knows what it doesn’t know”. Many deep learning models are miscalibrated in the sense that they predict the wrong label with very high confidence, which makes them difficult to trust. For a good introduction (including a discussion of ECE), see [2].
>
> References
> 1. Hinton, G., Vinyals, O., & Dean, J. (2015). Distilling the knowledge in a neural network. arXiv preprint arXiv:1503.02531.
> 2. Guo, C., Pleiss, G., Sun, Y., & Weinberger, K. Q. (2017, July). On calibration of modern neural networks. In International Conference on Machine Learning (pp. 1321-1330). PMLR.

---

### Official Review · Reviewer_xYbr · 2021-07-16

**Rating:** 7
**Confidence:** 3

**Summary:**

This paper interprets the knowledge distillation method from two aspects: generalization and fidelity. The key finds are:(1) Good student accuracy does not imply good distillation fidelity; (2) Student fidelity is correlated with calibration when distilling ensembles;(3) Optimization is challenging in knowledge distillation;(4) There is a trade-off between optimization complexity and distillation data quality.

**Limitations And Societal Impact:**

This paper did not address the limitations and potential negative societal impact. Please discuss any potential issues, welcoming a broader discussion that engages the whole community.

**Main Review:**

Strengths: This paper makes attempts to analyse whether knowledge distillation works. The experiments are mainly down on CIFAR100 with interesting findings.
Weaknesses: Multiple strategies about knowledge distillation were developed It is unclear if the same holds true for the rest especially that newer methods were shown to significantly outperform vanilla KD this makes the conclusions limited in scope.

Post-rebuttal: Thanks to the authors for addressing my concerns and questions. I tend to accept this work.

**Time Spent Reviewing:**

1hour

---

> ### Author Response · Authors · 2021-08-10
> **Individual Author Response: xYbr**
>
> Thank you for your time and feedback! In addition to this response, please see the separate post with our general response, which includes new ImageNet, IMDB, and initialization results. We hope you can consider raising your score in light of our responses, clarifications, and experiments inspired by your questions.
>
> 1. Why only ‘vanilla’ KD? We agree it is possible that our results do not extend to, for example, feature distillation methods. We chose to focus on ‘vanilla’ KD because it is the most natural choice when distilling ensembles, which was the original motivation for the project, and the most widely used KD approach. More sophisticated distillation methods are usually designed for a single teacher model and come laden with additional design choices that make analysis all the more difficult.
>
> 2. Social Impacts and Limitations: we address the social impacts and limitations of our work in Appendix D. We think that this work should have a positive social impact: for deploying deep learning in the real-world it is crucial to investigate how our procedures operate. Moreover, as we mention in section 4.3, distillation is being used for crucial healthcare applications like modelling covid-19. If the student is in fact a low-fidelity representation of the teacher, it could be dangerous to interpret its structure in such settings.

---

### Official Review · Reviewer_n5i1 · 2021-07-16

**Rating:** 7
**Confidence:** 4

**Summary:**

In the present work, the authors present some interesting phenomena observed when training a student NN architecture to mimic a teacher classifier through knowledge distillation. Instead of focusing on test performance (which is indeed the ultimate aim of this transfer learning paradigm) as in most previous work, they look at the student fidelity, i.e. its ability to actually reproduce the teacher behavior both on new data and on the distillation data (used to train the student). The authors highlight a low degree of fidelity observed in many learning scenarios and investigate its causes, showing that the optimization problem associated with knowledge distillation is hard and often solved pretty poorly. Another finding is that data augmentation procedures might help improve generalization performance or fidelity, but the two properties don't always come hand in hand.

**Limitations And Societal Impact:**

Limitations and social impact were only addressed in the supplementary materials.
I think an important limitation that could be discussed a bit more thoroughly is the difficulty of disentagling the observations presented in the paper and the many modeling choices made by the authors (training algorithm, hyperparameters, network architecture and dataset), given the quite different behavior observed with the Lenet setup on MNIST. While some of these aspects were somewhat explored in this work, I don't think the evidence is conclusive on the generality of the described phenomena.
Personally, I found the social impact discussion satisfactory.

**Main Review:**

I think the main findings reported in this paper, on the relationship between generalization performance and student fidelity yielded by distillation procedures, are quite unexpected and interesting. Many of the complex phenomena of deep learning are linked to a mismatch between training objectives and performance evaluation metrics, and understanding their interplay is key for making progress in the field.
On the practical side, I think this study could help motivate some further work on how to better optimize the knowledge distillation loss.

A few comments:
- I think the presentation is generally ok, but I found some of the figures quite hard to read (ex. fig 11). Sometimes maybe a table would convey the results more efficiently. Moreover many results from the appendix are referenced but not explained too well in the main text.
- The calibration error was not given too much space in the analysis, at least in the main text. However, this is the quantity that is more directly optimized through knowledge distillation. E.g. it is not clear in the experiments of figure 5 whether the degraded agreement can also be seen at the level of the KD.
- The comparison between students of independent teachers is referenced but not presented in a convincing way. I think this is an important point.
- I think the distillation temperature could play an important role in the results in fig. 5. Is the same behavior observed also with different temperatures? In the same figure, right plot, how many samples were added through augmentation?
- What happens if the distillation temperature is annealed? Could it help avoid the optimization problems observed in section 6?
- In fig.5, it seems that the reported agreement on the training set of the CombAugs experiments is lower than the test agreement in fig.3. How is this possible?
- Some statements in the paper are a bit too general for the presented data, e.g. "the drop in train agreement is even more pronounced when we use extensive data augmentation". But with the baseline augmentation, 100% agreement seems to be achievable.
- I think the evidence on the fact that adding more training epochs cannot be sufficient for achieving a perfect agreement is not conclusive. Particularly in the case of ResNets, some symmetry-breaking transitions could be needed in order to fall into the teacher basin of attraction and these could take longer times. This scenario is also compatible with the observations on the impact of the initialization. Maybe an interesting experiment would be to look at the effect of initializing the student close to the initial condition of the teacher (i.e. the untrained teacher).
- In the data recycling hypothesis paragraph, I don't understand why violating the i.i.d. assumption (which is not really valid anyway) would be a problem. Moreover, I think the fact that s_0 attains better generalization is to be expected since the teacher representations with the D_0 dataset induces correct labeling.

All in all, I think this work is interesting, but some additional experiments could help clarify the studied phenomena and support the very strong claims made in the discussion.


**Time Spent Reviewing:**

12

---

> ### Author Response · Authors · 2021-08-10
> **Individual Author Response: n5i1**
>
> Thank you for your time and feedback! We also found many of our experimental results quite counter-intuitive. In addition to these responses, please also see our separate general post, which includes new ImageNet, IMDB, and initialization experiments, inspired by your comments. We hope you can consider our responses in your final assessment.
>
> 1. Presentation: we agree that Figure 11 is rather difficult to parse at a glance, and will include a tabular version in the updated version. We will also review how the results in the appendices are introduced in the main text to improve the reader’s experience.
>
> 2. Calibration error: As discussed in section 3.1, the distillation loss is the tempered cross-entropy from teacher to student, which is equivalent to the KL divergence from teacher to student when the temperature is 1. For that reason in the main text we focus more on the test KL than the calibration error. If space permits, we will expand the discussion of ECE in the main text.
>
> 3. Independent teacher comparison: we explain the motivation behind the experiment in Section 3.2. We will review the presentation of the results for greater clarity.
>
> 4. Distillation temperature: We explored temperatures ranging from 0.5 to 16. Because earlier work has addressed the importance of temperature [1], we simply have presented the results for the best temperature we found (\tau=4). We included one temperature ablation in Figure 4, bottom row. If the temperature is not high enough, then the teacher labels are not very soft, and there is little benefit to distillation. However, we also found that increasing the temperature makes the optimization problem more difficult, amplifying the effect in Figure 5. The choice of optimal temperature ends up being similar to the choice of optimal distillation data, in that the goal is to produce soft teacher labels that can still be learned easily.
>
> 5. How many samples? It is difficult to estimate exactly how many samples the student can be said to have seen, since augmentations like color jitter are continuous. Because new augmentations are sampled every minibatch, technically the student has seen 15M different images (50K * 300 epochs), but obviously it is not the same as seeing 15M independent samples from the true data distribution.
>
> 6. Temperature annealing: Although we did not try annealing the distillation temperature directly, we tried a similar idea. Instead of the standard KD loss, we tried the convex combination of the cross-entropy between the teacher top-1 labels and the student labels, and the cross-entropy between the teacher soft labels and the student labels. The coefficient was linearly decayed throughout the course of distillation, with the hope that using soft labels less at the beginning of distillation would improve the result. One could achieve a similar effect by gradually increasing the temperature of the KD loss. Unfortunately the result of our experiment did not indicate any significant improvement.
>
> 7. Train vs. test agreement: because the test data is not augmented, it is possible for a student to have higher test agreement than train agreement because the augmented train data is being drawn from a distribution with greater support. We will clarify.
>
> 8. 100% agreement for baseline augmentation: It is very important to note that the teacher was also trained with the Baseline augmentation policy, which is simply the standard crops and flips that are almost always used for CIFAR. The teacher labels are essentially unaffected by the baseline augmentations, which is decidedly not the case for more aggressive augmentation policies.
>
> 9. Initializing with the initial teacher weights: thanks for the suggestion! We agree it’s an interesting experiment, and inspired by your comments, we’ve included the results in our general response (see separate post). To summarize, initializing the student with the teacher weights produces no functional improvement to fidelity, but it does produce students that are slightly closer in weight space (as measured by CKA [2]). Even if it did help, it would be difficult to extend the procedure to ensemble distillation.
>
> 10. On data recycling: One of the surprising things about the results in Figure 5 is that even though the student distilled on D_1 has significantly better test KL and test NLL, it has substantially worse accuracy. This is one of the fundamental examples in which doing ‘better’ at distillation can possibly cause your student to be less accurate! Fortunately the fix in that particular experiment is simple. Simply combine the teacher data and the new, unlabeled data together when doing distillation.
>
> 11. Limitations: we agree with the importance of disentangling design decisions, and we make an effort to control for many decisions (architecture, capacity, data augmentation, temperature, the use of unlabeled data, etc.). But there are of course always more possibilities, especially in deep learning. We hope that our response and additional experiments have helped convince you of the validity of our findings.
>
> References
> 1. Cho, J. H., & Hariharan, B. (2019). On the efficacy of knowledge distillation. In Proceedings of the IEEE/CVF International Conference on Computer Vision (pp. 4794-4802).
> 2. Kornblith, S., Norouzi, M., Lee, H., & Hinton, G. (2019, May). Similarity of neural network representations revisited. In International Conference on Machine Learning (pp. 3519-3529). PMLR.

---

> > ### Comment · Reviewer_n5i1 · 2021-09-01
> > **post-rebuttal feedback**
> >
> > I would like to thank the authors for carefully trying to address the many points raised in the reviews. I think that the additions to the manuscript help delineate the somewhat complex but intriguing interplay between fidelity and generalization. Personally, I see as a positive the fact that the whole phenomenon is pretty new and that it doesn't feel completely well understood, even after reading the paper. For me, this is good motivation for future work.
> >
> > I think this is a nice work that proposes some original insights and that it would be nice to see it accepted in Neurips. I will personally increase my score from 6 to 7: Good paper.

---

### Official Review · Reviewer_4uZD · 2021-07-16

**Rating:** 5
**Confidence:** 3

**Summary:**

The paper addresses the importance of student fidelity to the teacher in knowledge distillation. The major contribution is that the authors claimed that the fidelity is not related to accuracy, but correlates with calibration. And most previous knowledge distillation methods pay much attention to accuracy instead of fidelity, which is helpful for student generalization. These previous knowledge distillation methods also tested different ways of modifying data, changing the training procedure but unable to reach a high enough fidelity, potentially harming model generalization. The title of the paper is eye-catching, and its content seems inspiring.

**Limitations And Societal Impact:**

Yes. The authors provide adequate discussion of limitations and potential negative societal impact.

**Main Review:**

Strengths:

+ It is interesting that this paper focuses on the fidelity of the student network. The authors did plenty of experiments to conclude that fidelity is not related with accuracy, and found the factor to reach a high student fidelity.

+ The logic of this paper is quite clear, which makes it easy to read and understand.

+ There is a detailed analysis provided in the experimental section, which is solid support of the augments of this paper.

Weaknesses:

- It would be more convincing if some larger and more natural datasets such as ImageNet can be included in the experiment since fidelity is highly related to generalization.

- The discussion of related works is not comprehensive enough. Since the major claim of this paper is to verify if knowledge distillation really works, there should be more related literature included for discussion.

Other Comments:

1. The basic assumption of this paper is that great fidelity is beneficial for generalization which is not the same as accuracy. Here I have a side question, if the student well fits the teacher given some inputs, is it any overfitting problem as the input dataset cannot fully cover the whole space?

2. I am still not quite convinced by the main claim of this paper. The criteria of KD should be the the test accuracy of student model. I hope to see more explanation about this point in the rebuttal.


It is quite an interesting paper in this field. I recommend the score 5 due to the major weakness on evaluation datasets, of which I expect to see a larger one. I will consider upgrading the score if the authors provide a strong rebuttal.


**Time Spent Reviewing:**

4

---

> ### Author Response · Authors · 2021-08-10
> **Individual Author Response: 4uZD**
>
> Thank you for your time and feedback! In addition to this response, please see the separate post with our general response, which includes new ImageNet, IMDB, and initialization results. We hope you can consider raising your score in light of our responses, clarifications, and experiments inspired by your questions.
>
> 1. To clarify, our main claim is that it is very difficult to ‘succeed’ at solving the KD optimization problem (in the sense of finding a global optimum). Furthermore, we claim that many criteria, including accuracy, particularly when the teacher is an ensemble or a much larger model than the student, would be improved if we were able to attain higher fidelity. Therefore it is worth understanding the causes of low fidelity. We do not say the fidelity and accuracy are unrelated, rather they are not perfectly correlated.
>
> 2. Scope of the results: given the feedback from you and the other reviewers expressing desire to see similar results on a wider range of datasets, we have included new results on ImageNet and IMBD (text classification).
>
> 3. Literature review: although distillation is a broadly studied topic, surprisingly little of the literature addresses the topic of fidelity specifically, which is a significant point of novelty for our paper. This focus on fidelity may account for any omission of references on distillation more broadly, but we are happy to include any additional references.
>
> 4. Overfitting: if the student is overfitting to the teacher on the distillation dataset, we should be able to reduce the overfitting by adding more data. In practice adding more data does not make a big difference, so overfitting is unlikely to be the problem.

---

> > ### Comment · Reviewer_4uZD · 2021-08-27
> > **Post-rebuttal Discussion**
> >
> > Dear Authors,
> >
> > Thanks for your detailed response. I am glad to see the results on ImageNet, which are ensembles distillation and self-distillation. The major concern is that I have not seen much useful information on such a table. The only conclusion is more ensembles can improve the fidelity instead of test accuracy and the gap is enlarged during ensembles. I am slightly confused by the main claim of this paper. The relationship between fidelity and accuracy seems a case-by-case phenomenon (i.e., the acc on imagenet is not improved along with the increasing fidelity, and the IMDB is opposite). So, the KD fidelity cannot provide a comprehensive explanation for different scenarios, which makes its contribution somewhat weak. For the overfitting issue, I am confused about the author's response. Since adding more data does not make a big difference, how to address the problem of overfitting due to the high fidelity? As noticed in Fig. 1 of the paper, more generated data make the testing accuracy dropping and fidelity increasing. The student model tends to overfit the teacher due to the high fidelity, which degenerates its generalization. In summary, I think the overall quality of this paper is slightly below the NeurIPS, and I will keep my initial score.

---

> > > ### Author Response · Authors · 2021-08-27
> > > **Follow-up Response to Reviewer 4uZD**
> > >
> > > Thanks for engaging in further discussion! We would like to provide some important clarifications.
> > >
> > > “The only conclusion is more ensembles can improve the fidelity”: We believe there are more conclusions that can be drawn from the new ImageNet and IMDB results.
> > > 1) low fidelity is observed across a range of datasets of varying size and difficulty (not just CIFAR-100), which supports the scope of the claims in the paper. The ImageNet results are aligned with our CIFAR results, and complement these experiments.
> > > 2) the students with the highest test accuracy also have the highest test fidelity, demonstrating that when the teacher is a large model, improving fidelity can improve generalization.
> > >
> > >
> > > “The relationship between fidelity and accuracy seems a case-by-case phenomenon”: At first glance the relationship may appear irregular, but upon closer inspection the behavior is quite consistent. The key variable is the gap between the teacher accuracy and the student accuracy if both networks are trained in the standard way. In self-distillation there is no gap (because the networks are identical), so by necessity fidelity cannot improve student generalization beyond the teacher. When the teacher is large, improving test fidelity generally improves generalization, all else equal. Finally, some changes to distillation (e.g. using only unlabeled data for distillation) slightly improve test fidelity at a huge cost to train fidelity, resulting in a net negative effect on the final test accuracy.
> > >
> > >
> > >
> > > “How to address the problem of overfitting due to the high fidelity?”  In general the issue is not overfitting, it is underfitting, as shown in Figure 5. Self distillation is the only case where overfitting might be said to be a problem, and in that setting standard KD works very well (in the sense that students generalize well).
> > >
> > > General remark:
> > >
> > >
> > > We believe that investigations to understand popular procedures in deep learning are of paramount significance. Our work is the first to explicitly decouple distillation fidelity from generalization. We perform a remarkably large number of controlled experiments, and have several consistent and important practical takeaways. We have also now conducted ImageNet and IMDB experiments, which, as we clarify, help further strengthen and complement the already exceptionally extensive experiments in this work. There are several key points in our work, including the important observation that there is often a large fidelity gap between student and teacher, which is the explicit objective used in knowledge distillation. This result stands in contrast to general deep learning, where we can reliably achieve almost no training loss. We also clearly identify challenges in optimization (rather than student capacity, dataset size, etc), as the reason behind this observation, despite the optimization problem being very well solved in standard deep learning. We also note that other reviewers do see the value in our work, and support acceptance. We appreciate your questions, and hope you can reconsider based on our clarifications above. The practical takeaways, which we believe are important and consistent across our experiments, include:
> > >
> > >
> > > 1. You can improve the fidelity and accuracy of your student -- counterintuitively -- by making the teacher bigger and by ensembling several networks together.
> > > 2. The choice of distillation dataset is somewhat similar to the choice of temperature. Choosing images far from the training data will broaden the support of the dataset, but will make the average entropy of the teacher labels higher, and thus more difficult to optimize.
> > > 3. The best choice of distillation data is a combination of recycled teacher training data (low entropy labels, easy to fit) and new, in-distribution data (harder to fit, but improves identifiability).
> > > 4. If extra data is not available, additional data augmentation may help accuracy and fidelity, with similar caveats as the previous points. The augmented images should be somewhat similar to the training data (noise and unrelated images don’t help much). Expect longer training time/more sophisticated optimization procedures.
> > > 5. In particular, MixUp appears to be a good tradeoff between dataset support and optimization difficulty, resulting in small gains to accuracy and fidelity (a finding supported by concurrent work [2]).
> > > 6. Training longer and increasing student capacity may produce small improvements to accuracy and fidelity.
> > > 7. If properties like calibration are important to pass on to the student, you may need to switch distillation procedures (e.g. explore feature distillation) or take a small hit in accuracy.

---

> ### Author Response · Authors · 2021-08-24
> **Checking in**
>
> Dear Reviewer 4uZD, we would be grateful if you can let us know whether our responses have helped address your questions. We performed several experiments, including a large ImageNet experiment, at your request.

---

### Author Response · Authors · 2021-08-10
**General Author Response**

We thank all of the reviewers for their feedback and support. This is a general response, addressed to all reviewers and ACs. We will also make individual replies to address specific reviewer concerns as separate posts. As noted in the reviews, knowledge distillation (KD) is broadly used, yet poorly understood. While empirical studies focused on conceptual understanding are hard to evaluate, they bring immense value to the community. To our knowledge, this work is the first systematic investigation into the fidelity of distilled networks. We have identified a relatively unknown phenomenon -- students simply do not match their teachers as one would expect. We are able to say with confidence that a primary cause for low fidelity is convergence to suboptimal local minima in the distillation loss surface, which is a significant new result. Our work also has many takeaways for deep learning practitioners, which we enumerate below. Inspired by reviewer comments, our response includes three additional experiments, including ImageNet, text classification (IMDB), and a different student initialization scheme. All experiments are discussed below. We hope you will consider raising your score in light of our response.

*Intention*: We want to highlight that we are not claiming “knowledge distillation doesn’t work”. We are providing a nuanced but important investigation into the ways in which knowledge distillation does and does not work. In particular, we show there is often a significant disparity between student and teacher predictions, which is the explicit objective being used to train the student. This result is in contrast to standard deep learning, where the loss is relatively well-optimized. This disparity has many practical consequences, as we outline below.

*Novelty*: We provide the first systematic study of the specific objective used for training in knowledge distillation, decoupling our understanding of student fidelity from generalization. This study in itself is highly novel, with most works focusing almost exclusively on student generalization, even if fidelity is sometimes briefly reported. Our investigation leads to several novel insights, including: (1) Good student accuracy does not imply good distillation fidelity; (2) Student fidelity is correlated with calibration when distilling ensembles; (3) Optimization is unusually challenging in knowledge distillation; (4) There is a trade-off between optimization complexity and distillation data quality.

*Significance*: In general, investigations to understand popular procedures in deep learning are of paramount significance. In this case we see that the actual objective used in knowledge distillation is optimized very poorly. As enumerated below, this investigation has many practical take-aways. It additionally provides a mechanism for future methodological innovation: identifying optimization as the culprit, one can focus on building specialized optimizers for distillation procedures. And indeed, while self-distillation may be an exception, high fidelity students are generally aligned with desirable behaviour, particularly when the teacher is much better than the student, or if we care about other properties, such as calibration. Furthermore, in scientific tasks we wish to distill powerful black box models into simpler interpretable models, for the purpose of scientific discovery. Having the student be very different from teachers in these settings is especially problematic, leading to false inferences about structure discovered by the black-box model.

Practical takeaways
1. A very practical thing you can do to improve the accuracy and fidelity of your student is -- counterintuitively -- make the teacher bigger by ensembling several networks together. Ensemble labels are both more accurate and easier for the student to emulate.
2. The choice of distillation dataset is somewhat similar to the choice of temperature. Choosing images far from the training data will broaden the support of the dataset, but will make the average entropy of the teacher labels higher, and thus more difficult to optimize.
3. The best choice of distillation data is a combination of recycled teacher training data (low entropy labels, easy to fit) and new, in-distribution data (harder to fit, but improves identifiability).
4. If extra data is not available, additional data augmentation may help accuracy and fidelity, with similar caveats as the previous points. The augmented images should be somewhat similar to the training data (noise and unrelated images don’t help much). Expect longer training time/more sophisticated optimization procedures.
5. In particular, MixUp appears to be a good tradeoff between dataset support and optimization difficulty, resulting in small gains to accuracy and fidelity (a finding supported by concurrent work [2]).
6. Training longer and increasing student capacity may produce small improvements to accuracy and fidelity.
7. If properties like calibration are important to pass on to the student, you may need to switch distillation procedures (e.g. explore feature distillation) or take a small hit in accuracy.

Additional experiments

**ImageNet**: Inspired by reviewer comments, we are happy to say we have run ResNet-50 distillation experiments on the large-scale ImageNet dataset. Below, we provide the results for distilling ensembles of 1, 3 and 5 ResNet-50 teachers into a single ResNet-50 model. For each setting, we report the results averaged over 3 independent runs. The results further validate our CIFAR-100 experiments. In particular, top-1 agreement is again in the 80% - 90% range, adding more ensemble components to the teacher improves student accuracy and fidelity, and both the accuracy and fidelity gap between teacher and student can be observed. We will provide details and additional ImageNet experiments in the camera-ready version of the paper.

| # Ens. Components | Teacher Test Acc. | Student Test Acc. | T-S Test Agree. |   T-S Test KL   |
|------------------:|:-----------------:|:-----------------:|:---------------:|:---------------:|
|                 1 |  0.7476 (0.0014)  |  0.7530 (0.0010)  | 0.8551 (0.0014) | 0.2168 (0.0022) |
|                 3 |  0.7641 (0.0010)  |  0.7545 (0.0010)  | 0.8778 (0.0009) | 0.1572 (0.0003) |
|                 5 |  0.7667 (0.0010)  |  0.7562 (0.0007)  | 0.8836 (0.0004) | 0.1416 (0.0014) |

-------------------------------------------------------------------------------------------------------------------------------

**IMDB**: to avoid potential concern that our findings are limited to image classification, we also demonstrate similar results when distilling text classifiers on IMDB sentiment analysis data. Similar to our ImageNet experiment, we distill 1, 3 and 5 2-layer bidirectional LSTM teachers into a single LSTM of the same architecture. In particular, note that like our other results, the teacher accuracy is strictly improving, whereas the student accuracy ceases to improve from 3 to 5 teacher ensemble components, which is associated with a similar lack of improvement in agreement.

| # Ens. Components | Teacher Test Acc. | Student Test Acc. |  T-S Test Agree. |   T-S Test KL   |
|------------------:|:-----------------:|:-----------------:|:----------------:|:---------------:|
|                 1 |  79.3613 (0.1315) |  80.3533 (0.1984) | 86.4880 (0.5207) | 0.1242 (0.0124) |
|                 3 |  81.8067 (0.1286) |  81.1293 (0.0566) | 89.8320 (0.3489) | 0.0636 (0.0026) |
|                 5 |  82.2160 (0.2074) |  81.1667 (0.1956) | 90.7933 (0.1798) | 0.0519 (0.0007) |

-------------------------------------------------------------------------------------------------------------------------------

**Student Initialization**: we have also extended the experiment in Figure 6(b), examining the self-distillation regime again, initializing the student with the initial teacher weights. In this experiment we report the same metrics as the rest of the paper, with the addition of CKA [1]. We report the CKA for the preactivations of each of the three stages of our ResNet networks. There is a small increase in CKA, indicating that the teacher-initialized students are closer to their teachers in activation space, but functionally they are identical to their randomly initialized counterparts -- there is no appreciable change in accuracy, agreement, or predictive KL when compared to random initialization. We report the mean \pm st. deviation, estimated from 10 trials. The average teacher accuracy was 70.522 (0.412).

| Student Init. | Student Test Acc. | T-S Test Agree. |  T-S Test KL  | Test CKA (1)  | Test CKA (2)  | Test CKA (3)  |
|--------------:|:-----------------:|:---------------:|:-------------:|---------------|---------------|---------------|
|        Random |   73.721 (0.154)  |  77.174 (0.352) | 0.836 (0.016) | 0.939 (0.017) | 0.925 (0.027) | 0.885 (0.011) |
| Teacher Init. |   73.759 (0.373)  |  77.098 (0.238) | 0.838 (0.020) | 0.951 (0.017) | 0.937 (0.020) | 0.890 (0.015) |


References
1. Kornblith, S., Norouzi, M., Lee, H., & Hinton, G. (2019, May). Similarity of neural network representations revisited. In International Conference on Machine Learning (pp. 3519-3529). PMLR.
2. Beyer, L., Zhai, X., Royer, A., Markeeva, L., Anil, R., & Kolesnikov, A. (2021). Knowledge distillation: A good teacher is patient and consistent. arXiv preprint arXiv:2106.05237.

---

### Author Response · Authors · 2021-08-18
**We look forward to hearing from you**

Dear reviewers, with the discussion period underway, we're looking forward to your thoughts. We put a significant effort into our response, including a substantial ImageNet experiment. We hope you can consider our replies in your assessment. We're excited about the new discussion format at NeurIPS, and we look forward to hearing from you!

---

### Public Comment · ~Xinshao_Wang1 · 2022-10-16
**First, great work! Second, a dumb question, how do you define Knowledge Distillation?**

## LC and knowledge distillation (KD)
### (From the section \#6.2 of [ProSelfLC: Progressive Self Label Correction Towards A Low-Temperature Entropy State](https://arxiv.org/abs/2207.00118))
In the section 2.3, we have mathematically derived that some KD
methods [9, 27, 44, 83, 90] also modify labels. Therefore, LC and
KD are interchangeable in those cases. We use the term LC other
than KD mainly for two reasons: (1) LC is more descriptive; (2)
the scope of KD becomes much larger than label modification. For
example, when two models are trained, the consistency between
their predictions of a data point is rewarded in [5, 97], and a
large distance between their feature maps is penalised in [62].
Recently, multiple networks are trained for KD [17]. Regarding
self KD, the intraclass samples are constrained to have consistent
probability distributions [85, 91]. In another self KD [94], the
deepest classifier provides knowledge to supervise the shallower
classifiers. In a recent self KD method [90], Tf-KDself applies
two-stage training. In this work, we focus on improving the endto-end self LC. Therefore, some self KD methods [85, 91, 94],
maximising the consistency of different classifiers or intraclass
samples’ predictions, do not modify labels and are less relevant
for comparison. When it comes to the two-stage self LC method
[90], in our view, it can be an add-on, i.e., an enhancement plugin.
Therefore, exploiting ProSelfLC to improve non-self and stagewise LC approaches is an interesting area for future work.

---

### Decision · Program_Chairs · 2021-09-28

**Decision:**

Accept (Poster)

**Comment:**

The answer of the authors and subsequent discussion made most reviewers agree that this paper presents results about distillation that are solid and of interest to the community.

**Consistency Experiment:**

NeurIPS has a long history of experimentation. In 2014, NeurIPS ran an experiment in which 10% of submissions were reviewed by two independent committees to quantify the randomness in the review process. This year, we repeated a variant of this experiment to see how the quality of the review process has changed over time.  This paper was part of the experiment and was therefore assigned to two committees (consisting of reviewers, an Area Chair, and a Senior Area Chair) that reached independent decisions.  If both committees made the same recommendation, this recommendation was followed. If a single committee recommended acceptance, the paper was accepted (with the exception of a few cases in which the other committee identified what we considered a fatal flaw, e.g., an error in a key result).

Both committees reached the same decision: **Accept (Poster)**

The other committee assigned to the paper recommended **Accept (Poster)**.  You can find the other set of reviews, along with any follow up discussion with the authors here:
https://openreview.net/forum?id=Oa9RlXNggGy